# Stereotyped terminal axon branching of leg motor neurons mediated by IgSF proteins DIP-α and Dpr10

Lalanti Venkatasubramanian[1,2], Zhenhao Guo[1], Shuwa Xu[3], Liming Tan[3], Qi Xiao[3], Sonal Nagarkar-Jaiswal[4], Richard S Mann[2,5]*

[1]Department of Biological Sciences, Columbia University, New York, United States; [2]Department of Neuroscience, Mortimer B. Zuckerman Mind Brain Behavior Institute, New York, United States; [3]Department of Biological Chemistry, University of California, Los Angeles, Los Angeles, United States; [4]Department of Molecular and Human Genetics, Baylor College of Medicine, Houston, United States; [5]Department of Biochemistry and Molecular Biophysics, Columbia University, New York, United States

*For correspondence:
rsm10@columbia.edu

**Abstract** For animals to perform coordinated movements requires the precise organization of neural circuits controlling motor function. Motor neurons (MNs), key components of these circuits, project their axons from the central nervous system and form precise terminal branching patterns at specific muscles. Focusing on the *Drosophila* leg neuromuscular system, we show that the stereotyped terminal branching of a subset of MNs is mediated by interacting transmembrane Ig superfamily proteins DIP-α and Dpr10, present in MNs and target muscles, respectively. The DIP-α/Dpr10 interaction is needed only after MN axons reach the vicinity of their muscle targets. Live imaging suggests that precise terminal branching patterns are gradually established by DIP-α/Dpr10-dependent interactions between fine axon filopodia and developing muscles. Further, different leg MNs depend on the DIP-α and Dpr10 interaction to varying degrees that correlate with the morphological complexity of the MNs and their muscle targets.
DOI: https://doi.org/10.7554/eLife.42692.001

## Introduction

Animal behavior depends on the stereotyped morphologies of neurons and their assembly into complex neural circuits. Distinct neurons in many neural systems use combinations of effector molecules, such as cell-surface proteins, to form stereotyped connections with specific synaptic partners during circuit assembly (*Catela et al., 2015*; *Hong and Luo, 2014*; *Hattori et al., 2008*). Importantly, these effector molecules have specific roles in circuit assembly, ranging from pathfinding decisions to synapse formation, depending on the cellular context and developmental stage (*Peek et al., 2017*; *Koropouli and Kolodkin, 2014*; *Christensen et al., 2013*; *Sanes and Yamagata, 2009*).

The problem of circuit assembly is particularly important for motor circuits, where motor neurons (MNs) must form topographically organized connections between pre-motor interneurons in the central nervous system (CNS) and specific muscles in the periphery, thus establishing myotopic maps in both vertebrate and invertebrate systems (*Kania, 2014*; *Brierley et al., 2012*; *Baek and Mann, 2009*; *Landgraf et al., 2003*). Myotopic maps ensure that the correct inputs into MN dendrites are relayed through corresponding MN axons to the appropriate muscle groups (*Clark et al., 2018*; *Baek et al., 2017*; *Syed et al., 2016*). In order to assemble accurate myotopic maps, combinations of transcription factors specify distinct MN identities early during development, which in turn activate transcriptional programs to specify distinct MN morphologies during maturation, ranging from

the specification of distinct MN pools in the vertebrate spinal cord to individual MNs targeting the legs of adult *Drosophila* (*Enriquez et al., 2015*; *Santiago and Bashaw, 2014*; *Philippidou and Dasen, 2013*). Early work on MN axon pathfinding revealed that MN axons are capable of matching with their appropriate muscle targets even when their cell bodies are displaced along the A-P axis of the vertebrate spinal cord (*Landmesser, 2001*; *Hollyday and Hamburger, 1977*). Molecular evidence for synaptic matching between MNs and muscles was later identified in the form of attractive and repulsive receptor-ligand pairs expressed in subsets of MNs and muscles in both vertebrate and invertebrate systems (*Luria et al., 2008*; *Huber et al., 2005*; *Winberg et al., 1998*). Additionally there must be a balance between axon-axon and axon-muscle interactions to ensure the proper innervation and branching of MNs on their muscle targets (*Yu et al., 2000*; *Tang et al., 1994*; *Landmesser et al., 1988*). While much is known about the initial steps, in which MN axons navigate in response to guidance cues at several 'choice' points (*Bonanomi and Pfaff, 2010*; *Vactor et al., 1993*), less well understood is how MNs acquire and maintain their stereotyped terminal branching morphologies and thereby establish their synaptic connections known as neuromuscular junctions (NMJs).

The formation and maturation of NMJs is a highly precise process in which the terminal branches of each MN contain stereotyped numbers and sizes of synaptic connections (*Ferraro et al., 2012*; *Collins and DiAntonio, 2007*; *Johansen et al., 1989*). In vertebrates, differences in axon fasciculation and terminal branching morphologies are observed between MNs innervating 'fast' and 'slow' muscles, which have distinct physiological properties and functions (*Milner et al., 1998*). Further, the precise location of NMJ formation along each muscle fiber, defined by MN branch innervation as well as pre-patterned sites along each fiber, might also require reproducible terminal branching patterns (*Kummer et al., 2006*). This precision is also observed in *Drosophila* MNs that target larval body-wall muscles, where there are stereotyped differences between synapse size, terminal branching morphologies and electrophysiological properties (*Newman et al., 2017*; *Choi et al., 2004*; *Hoang and Chiba, 2001*).

In adult *Drosophila melanogaster*, ~50 morphologically unique MNs innervate 14 muscles in each leg. Each MN has stereotyped terminal branches that are located at specific regions of their muscle targets (*Brierley et al., 2012*; *Baek and Mann, 2009*; *Soler et al., 2004*). The similarities in the anatomical organization between *Drosophila* leg MNs and muscle fibers with their counterparts in the vertebrate limb suggest that common mechanisms might be involved. In order to identify genes used by *Drosophila* leg MNs, we characterized the expression patterns of various *Drosophila* cell-surface proteins in the adult leg neuromusculature using the MiMIC gene trap library (*Lee et al., 2018*; *Nagarkar-Jaiswal et al., 2015*; *Venken et al., 2011*). We focused on two families of genes that encode Ig-domain transmembrane proteins, the Dprs (Defective proboscis retraction) and DIPs (Dpr interacting proteins), which were identified as heterophilic binding partners (*Özkan et al., 2013*). Subsequent studies have shown that the DIPs and Dprs are expressed in specific neurons in the adult visual system in patterns that suggest they may be involved in mediating synaptic connectivity between 'partner' neurons (*Cosmanescu et al., 2018*; *Carrillo et al., 2015*; *Tan et al., 2015*). Additional functions of the DIPs and Dprs in axon self-adhesion in the olfactory system and synaptic specificity and synapse formation in the adult optic lobe and larval body-wall MNs have also been identified (*Xu et al., 2018a*; *Xu et al., 2018b*; *Barish et al., 2018*; *Cosmanescu et al., 2018*; *Carrillo et al., 2015*). Here we find that while *dprs* are broadly expressed in *Drosophila* adult leg MNs, the expression of *DIPs* tends to be more restricted to specific cell types, including small subsets of adult leg MNs. Most notably, DIP-α is expressed in a small number of adult leg MNs and its binding partner, Dpr10, is expressed in target leg muscles. Using in vivo live imaging of the leg MNs during development, we describe the process by which *Drosophila* leg MNs attain their unique axon targeting and terminal branching morphologies. Our results suggest that binding of DIP-α in MNs with Dpr10 in muscles is necessary for the establishment and maintenance of MN terminal branches in the adult leg. Moreover, the accompanying paper (*Ashley et al., 2018*) shows that the DIP-α-Dpr10 interaction plays a similar role in the larval neuromuscular system, suggesting a remarkably conserved function for these IgSF proteins at two stages of *Drosophila* development.

## Results

### Terminal branching of leg MNs occurs through sequential rounds of branching and defasciculation followed by synapse formation

To characterize the role of the DIP and Dpr proteins in MN development we first describe the process by which leg MN axons achieve their stereotyped muscle targeting and terminal branching patterns. We used a gene-trap within the *VGlut* locus to genetically label the glutamatergic leg MNs (*Figure 1A*) (*Diao et al., 2015*) and either an antibody or enhancer-trap for Mef2 (*Lin and Potter, 2016*), a transcription factor necessary for muscle development in *Drosophila* (*Lilly et al., 1995*), to label muscle precursors. Although we focused on the development of leg MNs targeting the foreleg (T1), the developmental processes described here are consistent across all three pairs of legs.

By the late third larval (L3) stage, adult leg MN axons from each thoracic hemisegment have exited the VNC through a single primary axon bundle and have targeted and terminate at the segment-specific ipsilateral leg imaginal disc, the precursor to the adult appendage (*Figure 1A*, *Figure 1—figure supplement 1A*). Larval MNs occupying the same nerve bundle are also labeled by *VGlut* but extend beyond the leg discs to target body wall muscles (*Figure 1—figure supplement 1B*). At this stage stereotyped groups of leg muscle precursor cells are present at specific positions in the leg imaginal disc (*Maqbool et al., 2006*). Shortly thereafter, 5 to 10 hr after puparium formation (APF), leg MN axon bundles begin to defasciculate and generate fine filopodia at their termini. By 20 hr APF, MN axons are organized into secondary bundles that target nascent muscles within each of four leg segments (Coxa, Trochanter, Femur and Tibia) (*Figure 1A*). Filopodia at the distal tips of these secondary bundles form net-like structures that insert between Mef2-expressing leg muscle precursor cells, the first indication that MNs are associated with distinct muscles (*Figure 1C*). By performing in vivo live-imaging on pupal legs expressing myr::GFP in the lineage that produces the largest number of leg MNs (LinA/15; 29 MNs) (*Enriquez et al., 2018*; *Brierley et al., 2012*; *Baek and Mann, 2009*), we observed that by 30 to 35 hr APF leg MN axons appear to maintain their initial connections to the same groups of muscle precursors even as their axons elongate and the shape of the leg disc changes (*Video 1*, *Figure 1A*). During this extension phase, as the main axon bundle lengthens, the process of axon branching continues to fine tune the targeting to distinct muscle fibers (*Video 2*). For example, at 25 hr APF the axons targeting the Tibia-long tendon muscle (Ti-ltm) remain fasciculated within the secondary bundle innervating the immature Ti-ltm (*Figure 1C*). Soon after leg extension is initiated, the Ti-ltm secondary bundle is sequentially split (*Figure 1D–E*) such that by 45 hr APF it has resolved into three distinct tertiary bundles with stereotyped terminal branching morphologies that are associated with distinct fibers of the Ti-ltm (*Figure 1E*). Although filopodia are still observed, they remain confined to the regions contacted by distinct terminal branches on each muscle fiber. From 45 to 60 hr APF, terminal branches maintain a similar branching pattern, while elaboration and pruning continues to establish finer branching, and finally each branch develops characteristic swellings known as synaptic boutons that ultimately mature into the NMJs present in the mature MNs of the adult (*Figure 1—figure supplement 1C–E*). Together, these observations indicate that MN axon bundles target distinct muscle groups as early as 20 hr APF, but stereotyped terminal branching onto specific muscle fibers is established between 25 to 45 hr APF.

### Expression of DIPs and dprs in distinct patterns in the *Drosophila* leg neuromuscular system

Because the establishment of stereotyped MN terminal branching involves the close association of developing MN axon termini with their target muscles, we expected cell-surface molecules to be required for this process. We focused on the Ig superfamily, the *dprs* and their interacting partners, the *DIPs*, and mapped the expression patterns of 8 *DIPs* and 16 *dprs* (*Carrillo et al., 2015*; *Özkan et al., 2013*) in the adult leg using MiMIC insertions converted to *T2A-Gal4* lines (*Lee et al., 2018*) (*Figure 2*, *Figure 2—figure supplement 1A*, *Supplementary file 1*). In general, the *dprs* are more widely expressed than the *DIPs*, as all the *dpr-MiMIC-T2A-Gal4* lines labeled the majority of adult leg MNs and leg sensory neurons (SNs) (*Figure 2A,E*, *Figure 2—figure supplement 1A–B*). In contrast, the *DIPs* were either expressed in a specific subset of leg MNs (DIP-α, DIP-β, DIP-ζ) (*Figure 2B*), in many but not all leg MNs (DIP-γ) (*Figure 2B*), or in subsets of leg MNs, SNs and/or

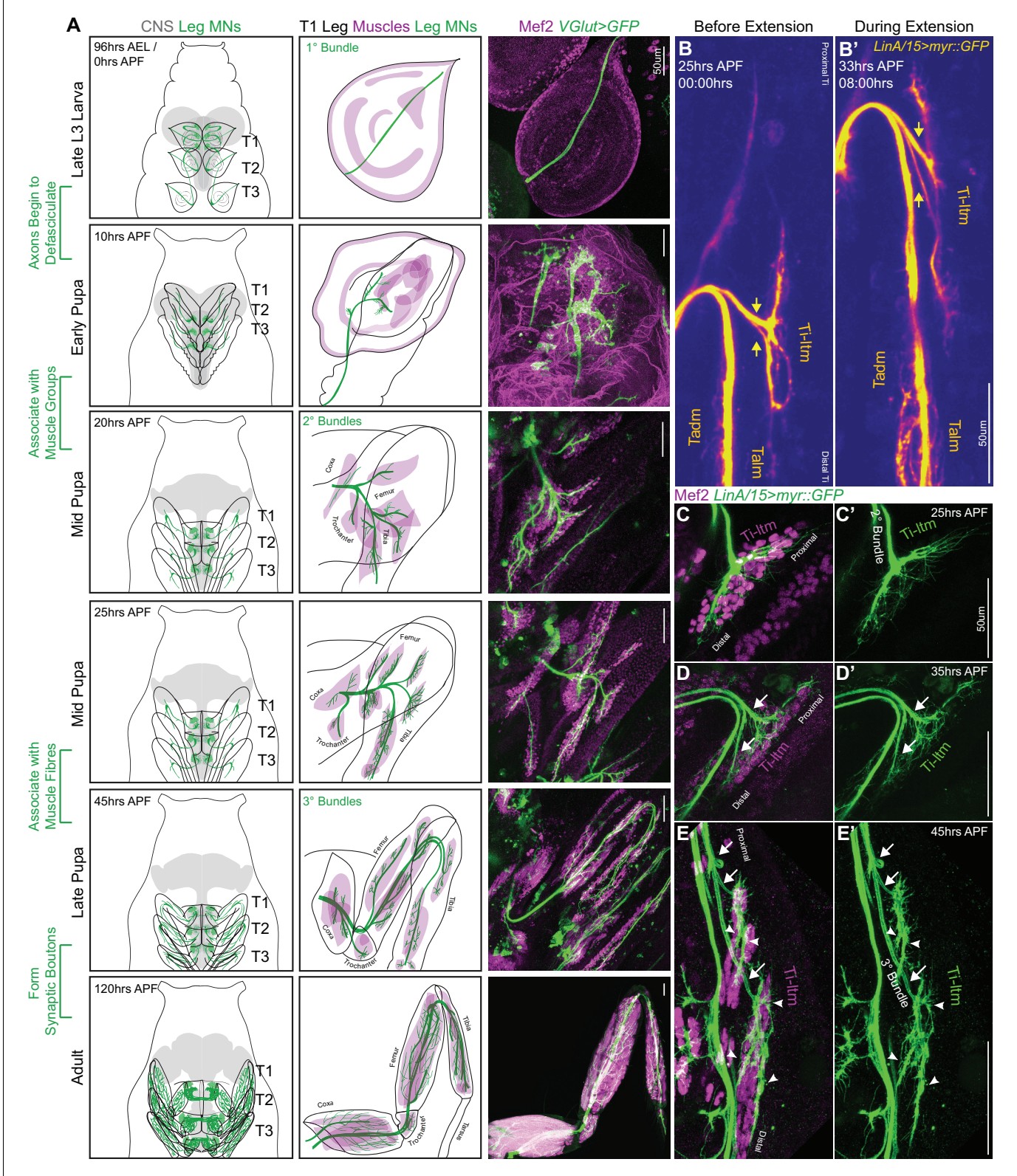

**Figure 1.** Sequential defasciculation and branching of developing *Drosophila* adult leg motor neurons. (**A**) Development of *Drosophila* adult leg motor neurons across six distinct time points during pupariation – Late L3 (96 hr AEL/0 hr APF), Early Pupa (10 hr APF), Mid Pupa (20 hr and 25 hr APF), Late Pupa (45 hr APF) and Adult (120 hr APF). Left Column: Schematic representation of *Drosophila* larval to adult stages denoting the locations of adult leg MN cell bodies and dendrites (green) in the CNS (gray) along with axons (green) targeting ipsilateral legs (T1 - forelegs, T2 - midlegs and T3 -

*Figure 1 continued on next page*

*Figure 1 continued*

hindlegs). Middle Column: Schematic representation of the developing T1 leg denoting the locations of muscle precursors (magenta) and leg MN axons (green). Locations of muscles within the four leg segments (Coxa, Trochanter, Femur and Tibia) are denoted from 20 hr APF onwards. Right Column: Leg MN axons in the developing T1 leg labeled by *VGlut-QF >10XUAS-6XGFP* (green) and stained for Mef2 (magenta) to label muscle precursors. Mature MNs and muscles in the Adult T1 leg are labeled using *OK371-Gal4 > 20XUAS-6XGFP* and *Mef2-QF > 10XQUAS-6XmCherry* respectively. (scale Bar: 50 μm) (B) Snapshots from a time-lapse series of developing LinA/15 leg MNs expressing myr::GFP at 25 hr APF (B); before extension) and 35 hr APF (B'); after extension) (see also *Video 1*). Arrows denote distinct axon bundles within the Ti-ltm-targeting bundle. Axon bundles are labeled according to muscle targeting – Ti-ltm: Tibia-long tendon muscle, Tadm: Tarsal depressor muscle, Talm: Tarsal levator muscle. (scale Bar: 50 μm) (C–E) Confocal images of LinA/15 Ti-ltm-targeting leg MN axons expressing myr::GFP (green) and muscles stained for Mef2 (magenta) at 25 hr APF (C–C'), 35 hr (D–D') and 45 hr APF (E–E'). Arrows point to defasciculating tertiary bundles and arrowheads (E–E') point to terminal axon branches. (scale Bar: 50 μm).

DOI: https://doi.org/10.7554/eLife.42692.002

The following figure supplement is available for figure 1:

**Figure supplement 1.** Sequential defasciculation and branching of developing *Drosophila* adult leg motor neurons.
DOI: https://doi.org/10.7554/eLife.42692.003

muscles (*DIP-δ, DIP-ε, DIP-η, DIP-θ*) (*Figure 2C–E*). The more widespread expression of the *dprs* is also observed in the *Drosophila* optic lobe, mushroom body and protocerebral bridge (*Davis et al., 2018*). Second, unlike the neurons projecting to the medulla neuropil in the visual system (*Tan et al., 2015*), *DIP* and *dpr* expression patterns in the leg were not selectively biased to either the pre/post-synaptic partner of the circuit as both *DIPs* and *dprs* are expressed in leg MNs, SNs and muscles (*Figure 2E*, *Figure 2—figure supplement 1B*). Interestingly, only *DIP-ε* and *dpr10* are expressed broadly in adult leg muscles (*dpr1* is expressed in a single fiber in the most proximal muscle of the Coxa), indicating that interactions between cognate DIP-Dpr pairs might function at multiple steps during the development of the adult leg, from axon fasciculation, synaptic specificity to proper synapse formation, consistent with their roles in the adult olfactory system, larval body-wall NMJ formation, as well as in the adult optic lobe (*Xu et al., 2018a*; *Xu et al., 2018b*; *Barish et al., 2018*; *Carrillo et al., 2015*).

## DIP-α is necessary for the terminal branching of three leg MNs

Since *dpr10* is strongly expressed in leg muscles, we initially focused on a potential role for its strongest binding interactor, *DIP-α*, in axon targeting (*Cosmanescu et al., 2018*). To examine *DIP-α*-expressing neurons at high resolution, we identified an enhancer from the *DIP-α* locus (*DIP-α-A8*) that specifically labels three of four adult *DIP-α* expressing leg MNs (*DIP-α-A8* also labels two rows of segmentally repeating larval MNs; *Figure 3A,B*, *Figure 3—figure supplement 1A–B*). Of the three adult leg MNs labeled by *DIP-α-A8*, two MNs target long tendon muscles (ltms), one in the Femur (αFe-ltm, which targets the Femur-long tendon muscle (Fe-ltm, also called ltm2)) and one in the Tibia (αTi-ltm, which targets the Tibia-long tendon muscle (Ti-ltm, also called ltm1)) (*Soler et al., 2004*). The third MN labeled by *DIP-α-A8*, αTi-tadm, targets the tarsal depressor muscle (tadm) located in the Tibia (*Figure 3A*). Based on their expression of DIP-α, we collectively refer to these three MNs as α-leg MNs.

We noticed a striking absence of terminal branching in α-leg MNs in homozygous *DIP-α* mutant animals using multiple alleles and genetic backgrounds (null, chromosomal deficiency, and homozygous *MiMIC-T2A-Gal4* – see Supplementary File 2) (*Figure 3B–D*). Similar defects were not observed in mutants for other *DIP*-expressing leg MNs, for example *DIP-β*, *DIP-γ* mutant or *DIP-ζ* knock-down animals (*Figure 3—figure supplement 1C*). Interestingly, the terminal branching of αFe-ltm, αTi-ltm and αTi-tadm displayed different but consistent penetrance of the mutant phenotype. αFe-ltm lost all terminal axon branches in 80–100% of the mutant samples analyzed while αTi-ltm lost all terminal axon branching in 20–40% of mutant samples analyzed (*Figure 3D*). The remaining αTi-ltm samples had some terminal branches, which were highly reduced in length and/or number (*Figure 3C,E*). αTi-tadm rarely displayed a complete loss of terminal axon branching (only in homozygous *DIP-α-T2A-Gal4* animals), but showed a loss of two to three terminal branches in several samples (*Figure 3C–E*). Strikingly, even when αTi-ltm and αFe-ltm have no terminal branches, their axons enter the leg and reach the vicinity of their muscle targets (*Figure 3—figure supplement 1D*). These results suggest that *DIP-α* is not required for these MNs to reach their respective muscle targets but

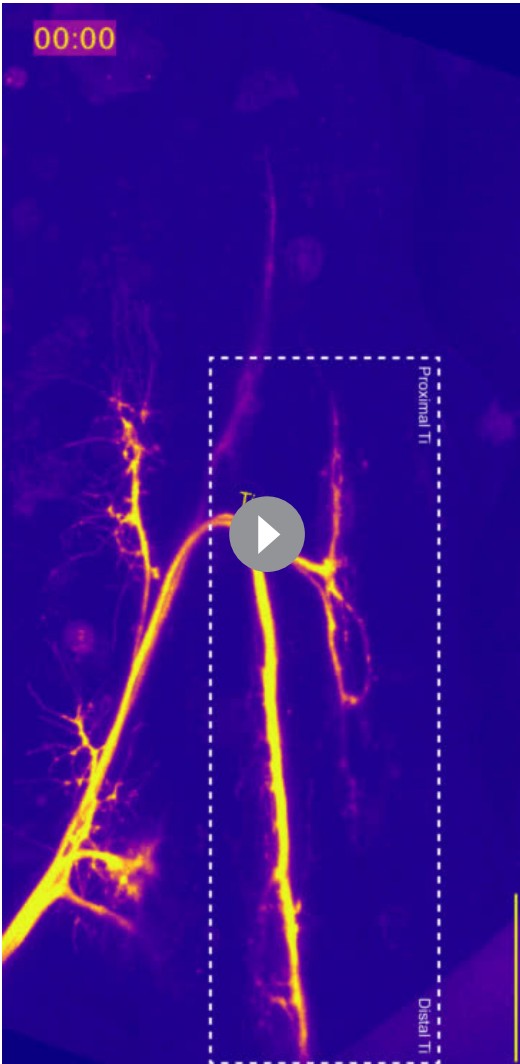

**Video 1.** Live imaging of developing LinA/15 leg MNs between ~25 - 37 hr APF. WT Time lapse in vivo live imaging of developing T1 LinA/15 leg MNs expressing myr::GFP (yellow) between ~25 hr APF (00:00) to ~37 hr APF (12:20) (10 min interval, five fps). In the first frame the Ti-ltm targeting secondary axon bundle is labeled within the white-dotted box demarcating the entire Ti segment. Leg extension is initiated at ~30 hr APF and axons within the secondary bundle begin to defasciculate while filopodial branches maintain physical contact with their muscle targets (*Figure 1B–D*, *Figure 1—figure supplement 1C–E*). (scale bar: 50 µm).
DOI: https://doi.org/10.7554/eLife.42692.004

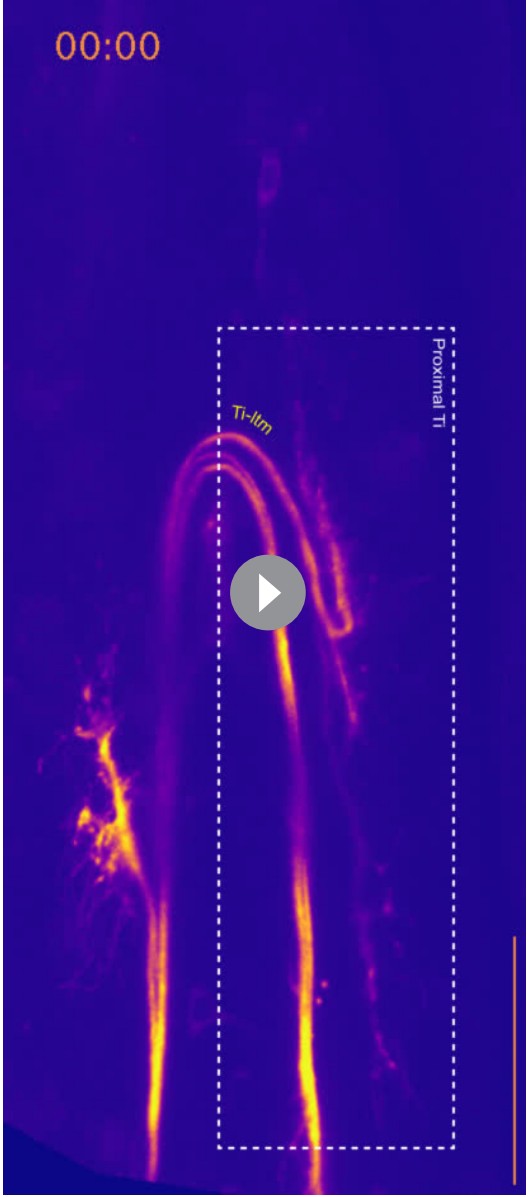

**Video 2.** Live imaging of developing LinA/15 Ti-ltm targeting leg MNs between ~40 - 50 hr APF. WT Time lapse in vivo live imaging of developing T1 LinA/15 Ti-ltm targeting leg MNs expressing myr::GFP (yellow) between ~40 hr APF (00:00) to ~50 hr APF (12:20) (10 min interval, five fps). In the first frame the Ti-ltm targeting secondary axon bundle is labeled within the white-dotted box. The generation of stable terminal branches occurs between ~43 to 45 hr APF, at which point leg muscles are being reorganized into distinct muscle fibers (*Figure 1B–D*, *Figure 1—figure supplement 1C–E*). (scale bar: 50 µm).
DOI: https://doi.org/10.7554/eLife.42692.005

is required to generate their stereotyped terminal branching morphologies. Importantly, terminal branching was fully restored in the α-MNs when *DIP-α-A8-Gal4* was used to re-introduce DIP-α only in these leg MNs in a *DIP-α*

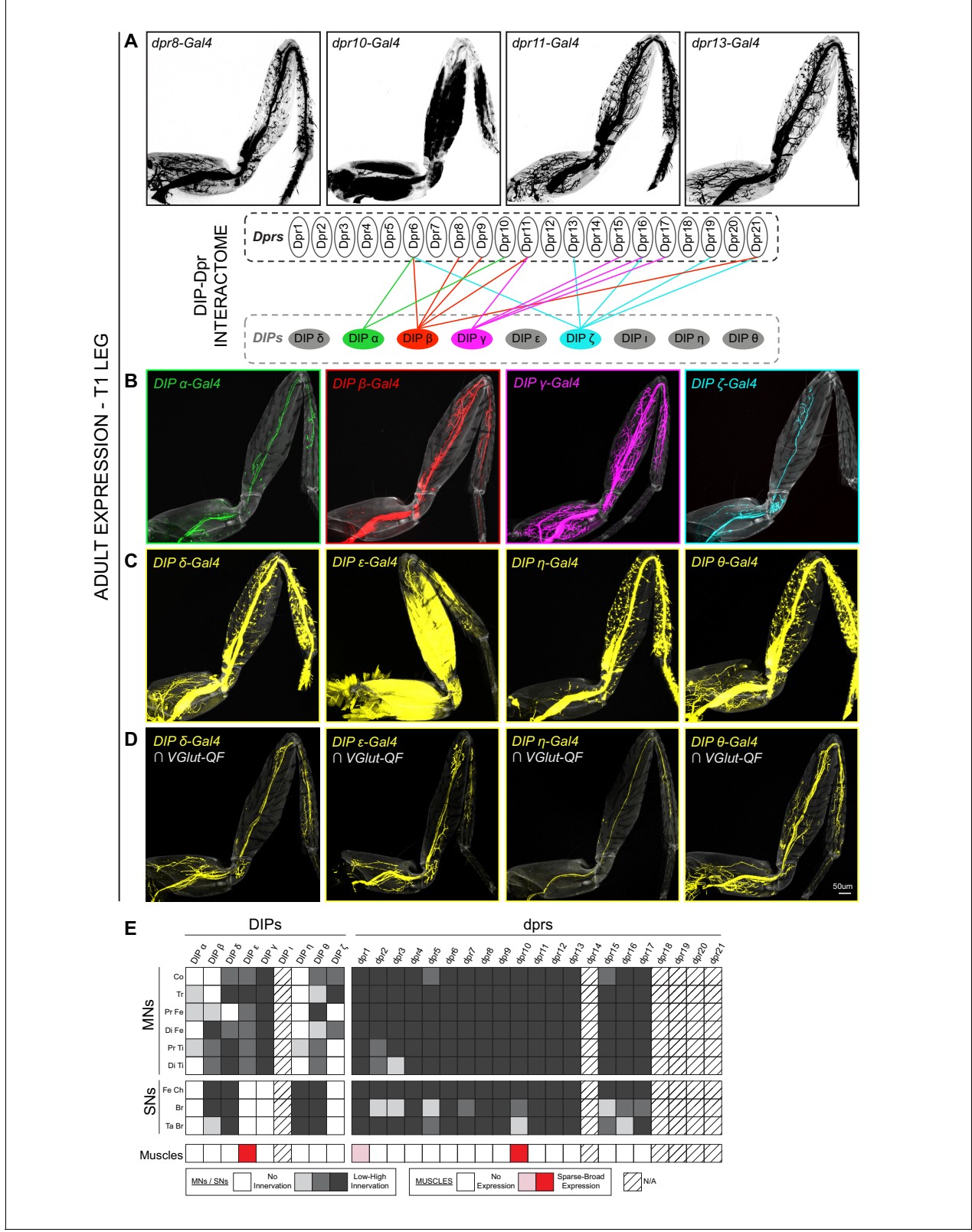

**Figure 2.** Expression patterns of *DIPs* and *dprs* in *Drosophila* T1 adult leg neuro-musculature. (**A–B**) *dpr* (**A**) and *DIP* (**B**) expression patterns in the *Drosophila* T1 adult leg for a subset of heterophilic binding partners identified by DIP-Dpr 'interactome' studies (*Carrillo et al., 2015*; *Özkan et al., 2013*): *DIP-α* (green) and *dpr10* (black); *DIP-β* (red) and *dpr8* (black); *DIP-γ* (magenta) and *dpr11* (black); *DIP-ζ* (cyan) and *dpr13* (black). These *DIPs* were selected because they are MN-specific in the legs. The expression patterns in this and other panels were generated with MiMIC Gal4 insertions (see
*Figure 2 continued on next page*

*Figure 2 continued*

*Supplementary file 1*). (**C**) Expression of four additional *DIPs* (*DIP-δ, DIP-ε, DIP-η*, and *DIP-θ*) in the T1 adult leg (yellow). In addition to MNs, these *DIPs* are expressed in leg sensory neurons (*DIP-δ, DIP-η*, and *DIP-θ*) or muscles (*DIP-ε*). (**D**) *DIP-δ, DIP-ε, DIP-η*, and *DIP-θ* expression restricted to glutamatergic MNs neurons in the T1 adult leg using a genetic intersectional approach (see Materials and methods). (scale bar: 50 μm). (**E**) Heat-map summary of *DIP-dpr* expression patterns in the T1 leg. Each column represents a distinct *DIP* or *dpr* expression pattern and each row represents a specific component of the adult leg-neuro-musculature. MN expression is categorized according to their terminal branching in different segments of the leg: Co, Coxa; Tr, Trochanter; Pr Fe, Proximal Femur; Di Fe, Distal Femur; Pr Ti, Proximal Tibia; Di Ti, Distal Tibia. SN expression is categorized according to their expression in sub-types of SNs (*Tuthill and Wilson, 2016*): Fe Ch, Femur Chordotonal Organ; Br, Bristle SNs; Ta Br, Tarsal Bristle SNs (campaniform sensilla and hairplate SNs were not included in the expression analysis). Muscle expression is not categorized because two of the three lines were broadly expressed in most muscles. (*) *dpr1* is expressed in a single muscle fiber entering the Coxa leg segment.

DOI: https://doi.org/10.7554/eLife.42692.006

The following figure supplement is available for figure 2:

**Figure supplement 1.** Expression patterns of additional *dprs* in the *Drosophila* T1 adult leg.

DOI: https://doi.org/10.7554/eLife.42692.007

mutant background (*Figure 3B,D*, *Figure 3—figure supplement 1E*).

In order to test whether DIP-α is sufficient to induce terminal branching at the ltms of a MN that normally does not target these muscles, we ectopically expressed DIP-α in LinB/24 leg MNs, which normally target muscles in the Coxa, Trochanter and distal Femur (*Enriquez et al., 2015*; *Brierley et al., 2012*; *Baek and Mann, 2009*), using MARCM (*Lee and Luo, 2001*) and the strong MN driver *VGlut(OK371)-Gal4* (*Mahr and Aberle, 2006*) (*Figure 3F*). We also performed this experiment in a *DIP-α* mutant background using an enhancer-trap (*hkb-Gal4*) expressed in LinB/24 MNs. In both cases normal targeting and terminal branching of Lin B/24 neurons was observed in nearly all samples; in only one of nine *DIP-α* mutant samples we observed branching at the Fe-ltm (*Figure 3F*). Because DIP-α was not able to efficiently target non-DIP-α-expressing leg MNs to the ltm, we hypothesize that rare ectopic branching events might be a consequence of stabilizing occassional 'stray' filopodia that come close to the ltm during pupal development. Additionally, we did not observe any obvious defects in dendritic arborization of the α-leg MNs in *DIP-α* mutants (*Figure 3— figure supplement 1F*).

From our expression analysis of the DIPs in the leg MNs, we also identified the expression of *DIP-β* in the α-ltm MNs (*Figure 3—figure supplement 2A*). In order to test for combinatorial DIP functions in leg MN targeting we assessed the function of DIP-β in these MNs. Loss of *DIP-β* alone did not affect the terminal branching of either α-ltm MN. Further, removing *DIP-β* in a *DIP-α* mutant background did not increase the penetrance or frequency of the terminal branching defects of αTi-ltm (*Figure 3—figure supplement 2B*) and expressing DIP-β in these neurons did not rescue the *DIP-α* mutant phenotype (*Figure 3—figure supplement 2C*). While these results do not rule out the possibility that DIP-β performs other functions in the α-ltm MNs, they confirm that *DIP-α* is primarily responsible for the terminal branching of the α-ltms described above.

## DIP-α can rescue terminal branching defects late in development

To further assess the role of DIP-α in terminal branching we characterized the spatial and temporal expression of DIP-α during pupal development. Using MARCM we first assigned the α-leg MNs to the LinA/15 adult leg MN lineage (*Baek and Mann, 2009*; *Brierley et al., 2012*) (*Figure 4A*, *Figure 4—figure supplement 1A*). By mapping the expression of *DIP-α* over the course of metamorphosis, we noticed that *DIP-α* turns 'ON' sequentially in the three LinA/15 α-leg MNs between 10 and 25 hr APF (*Figure 4A*). At 25 hr APF, the immature axons of all three α-leg MNs can be identified in the developing leg, within their respective secondary axon bundles and associated with their respective muscle groups (*Figure 4B*). In parallel, we used the MiMIC-GFP protein fusion (*DIP-α-GFSTF*) (*Nagarkar-Jaiswal et al., 2015*; *Tan et al., 2015*; *Carrillo et al., 2015*) to characterize the sub-cellular localization of DIP-α protein in the leg MNs during development (*Figure 4C*, *Figure 4— figure supplement 1B–D*). From the onset of expression until the adult, DIP-α is continually observed in the entire axon terminals of all three α-leg MNs. By 45 hr APF DIP-α localizes to the fine filopodial projections that are closely associated with the developing muscles and in the adult DIP-α is localized to the presynaptic sites of the mature synaptic boutons along the terminal branches (*Figure 4—figure supplement 1D*).

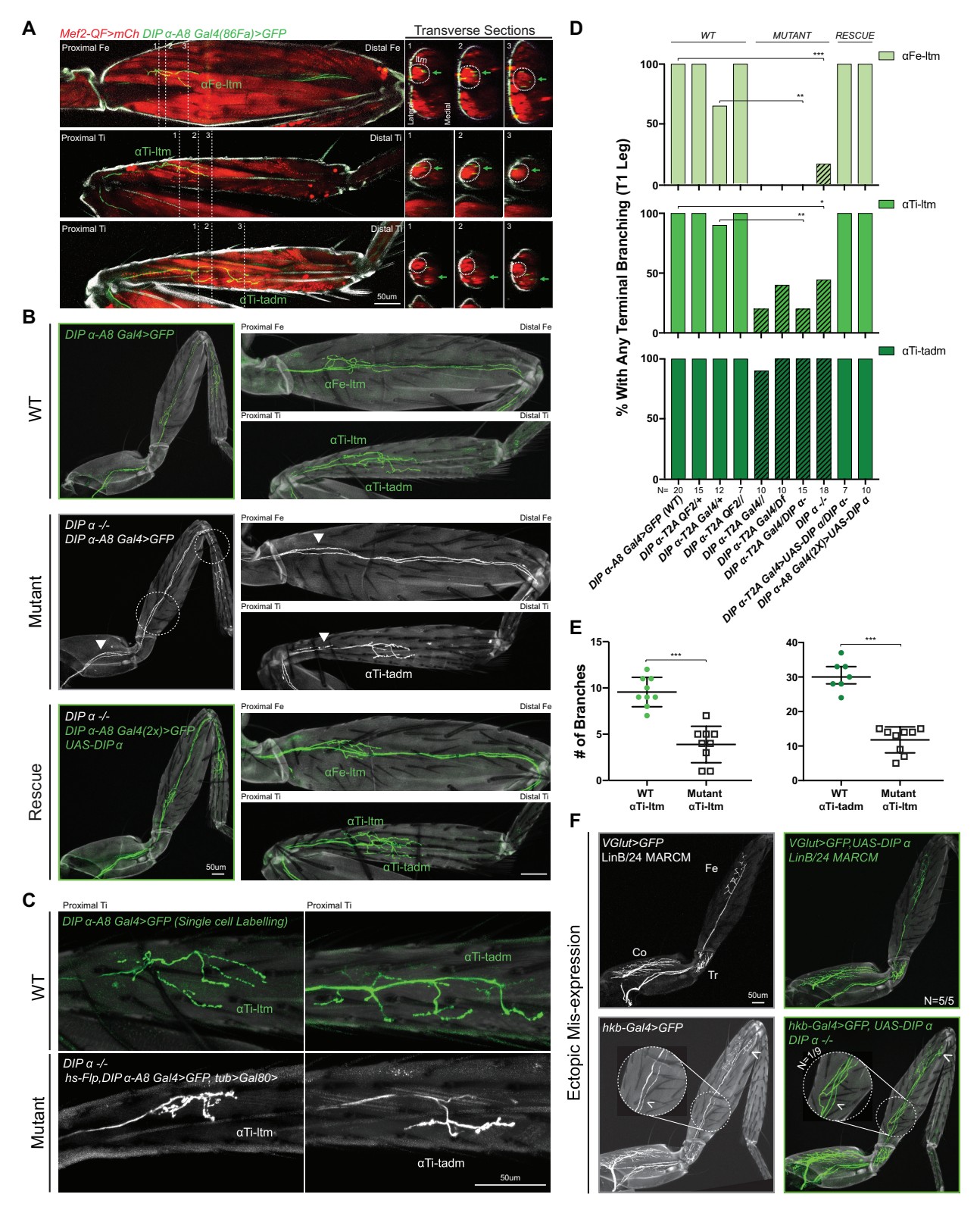

**Figure 3.** Effects of mutating DIP-α on the terminal branching of α-leg MNs. (**A**) Left Column: Proximal-Distal (P–D) oriented Fe and Ti T1 adult leg segments depicting axon muscle-targeting of three DIP-α expressing leg MNs labeled by *DIP-α-A8-Gal4(86Fa)>20XUAS-6XGFP* (green) (*Figure 3—figure supplement 1D*) (See Materials and methods). Muscles are labeled using *Mef2-QF > 10XQUAS-6XmCherry* (red); Grey; cuticle. MNs are named according to the muscle target (αFe-ltm, αTi-ltm, and αTi-tadm) (*Soler et al., 2004*). Right Columns: Transverse sections of Fe and Ti leg segments at

*Figure 3 continued on next page*

*Figure 3 continued*

specific locations along the P-D axis, corresponding to the numbered white dotted lines on the left, depicting terminal branching (green arrows) on the Fe and Tiltms (encircled by white dotted lines) and tadm. (scale bar: 50 µm). (**B**) Terminal branching of the T1 α-leg MNs labeled by *DIP-α-A8-Gal4 (86Fa)>20XUAS-6XGFP* in wild type (WT), DIP-α mutant and rescue contexts. Left; T1 legs; Right; Fe and Ti leg segments (axons; green (WT/rescue) or white (mutant), cuticle; grey). Absence of terminal branching of the α-ltm MNs in the DIP-α mutant T1 leg is indicated by white dotted circles; White arrowheads demarcate axons reaching the vicinity of their muscle targets (refer to *Figure 3—figure supplement 1D*). (scale bar: 50 µm). (**C**) Intermediate terminal branching defects in T1 legs displayed by αTi-ltm and αTi-tadm in *DIP-α* mutants. Single cell labeling of αTi-ltm and αTi-tadm terminal branches in the T1 proximal Ti is shown in WT (green) and *DIP-α* mutant (white). (scale bar: 50 µm). (**D**) Quantification of mutant phenotypes (αFe-ltm, light green; αTi-ltm, medium green; and αTi-tadm, dark green) in WT (N = 20), mutant (diagonal lines) and rescue contexts (N = 7 to 20) using a *DIP-α* null, chromosomal deficiency and *MiMIC-T2A-Gal4/QF* as indicated. Statistical significance was determined using Fisher's exact test: *p<0.05; **p<0.01; ***p<0.001 (**E**) Quantification of number of branches on αTi-ltm and αTi-tadm single-cell samples in WT and *DIP-α* mutant contexts using genotypes indicated in *Figure 1C*. Statistical significance was determined using a two-tailed unpaired t-test for αTi-ltm samples, where error bars represent mean ± SD and a Mann-Whitney U test for αTi-tadm samples, where error bars represent median ± interquartile ranges. ***p<0.001 (**F**) Ectopic expression of DIP-α in LinB/24 leg MNs targeting the Coxa, Trochanter and Distal Fe using *OK371-Gal4* MARCM (Top) or an enhancer trap *hkb-Gal4* (Bottom) which also labels an additional leg MN targeting the distal Fe (white arrowhead). Normal axon targeting of LinB/24 leg MNs (white) is shown on the left without any terminal branching at the Fe-ltm. However, in a rare case (N = 1/9), ectopic expression of DIP-α using *hkb-Gal4* in a *DIP-α* mutant background caused ectopic branching at the Fe-ltm (white arrowhead within magnified inset). (scale bar: 50 µm).

DOI: https://doi.org/10.7554/eLife.42692.008

The following figure supplements are available for figure 3:

**Figure supplement 1.** Characterization of *DIP-α* and other *DIP* mutants.

DOI: https://doi.org/10.7554/eLife.42692.009

**Figure supplement 2.** Co-expression and phenotypic analysis of *DIP-α* and *DIP-β* in adult leg MNs.

DOI: https://doi.org/10.7554/eLife.42692.010

Although these results reveal the timing and location of DIP-α expression, they do not tell us when DIP-α is required during development. To address this question we conducted a temporal rescue of the terminal branching phenotype using a temperature-sensitive Gal80 to inhibit *DIP-α-T2A-Gal4* activation of DIP-α in a *DIP-α* mutant background (*Figure 4D–E*, *Figure 4—figure supplement 2A*). In parallel, we examined positive (no *tub-Gal80$^{ts}$*) and negative (no *UAS-DIP-α-V5*) control animals from the same cross to test for an effect of the temperature shift (*Supplementary file 2*). We focused specifically on the terminal branching of αFe-ltm since the targeting of αTi-ltm was partially affected by the temperature shift even in the positive control (*Figure 4—figure supplement 2A*). Surprisingly, DIP-α is able to rescue the terminal branching in 90% of mutant αFe-ltm samples when expressed as late as 75 hr APF. Even when provided at 125 hr APF, which coincides with eclosion, partial rescue was observed in 70% of mutant samples, although rescue at this time point consisted of shorter terminal branches compared to the positive control (*Figure 4D*).

## Dpr10 expression in muscles is necessary for terminal branching of the α-Leg MNs

From the interactome measurements of the DIPs and Dprs, DIP-α interacts exclusively with Dpr6 and Dpr10 (*Cosmanescu et al., 2018Carrillo et al., 2015*; *Özkan et al., 2013*). Dpr10, in turn, most strongly binds DIP-α while Dpr6 can interact with DIP-α, DIP-β, DIP-ε, and DIP-ζ (*Cosmanescu et al., 2018*). Using double and single mutants of *dpr6* and *dpr10* we found that *dpr10* alone was necessary for the terminal branching of the α-leg MNs: *dpr10* mutants reduced terminal branching of αFe-ltm and αTi-ltm from 100% in the control to ~9% and 36%, respectively (*Figure 5A,D*). Notably, the same trends in penetrance and frequency of the terminal branching phenotype were observed in αFe-ltm, αTi-ltm, and αTi-tadm for *dpr10* and *DIP-α* mutants. Because *dpr10* is also expressed in SNs and MNs we used RNAi to knockdown *dpr10* specifically in muscles using *Mef2-Gal4* and separately in MNs using *OK371-Gal4*, and only observed a terminal branching phenotype when *dpr10* was reduced in muscles (*Figure 5—figure supplement 1A–B*). The *dpr10* mutant phenotype was partially rescued by expressing Dpr10 in the muscles using *Mef2-Gal4* but, curiously, this manipulation induced patchy expression of *DIP-α-T2A-QF* in additional leg cells (*Figure 5B,D*, *Figure 5—figure supplement 1C*). As an additional test, rescuing *dpr10* expression using *DIP-ε-T2A-Gal4*, which is also expressed in leg muscles, in *dpr10* mutants significantly rescued the terminal branching phenotype of αFe-ltm to 85.7% compared to controls (*Figure 5D*, *Figure 5—figure supplement 1A*).

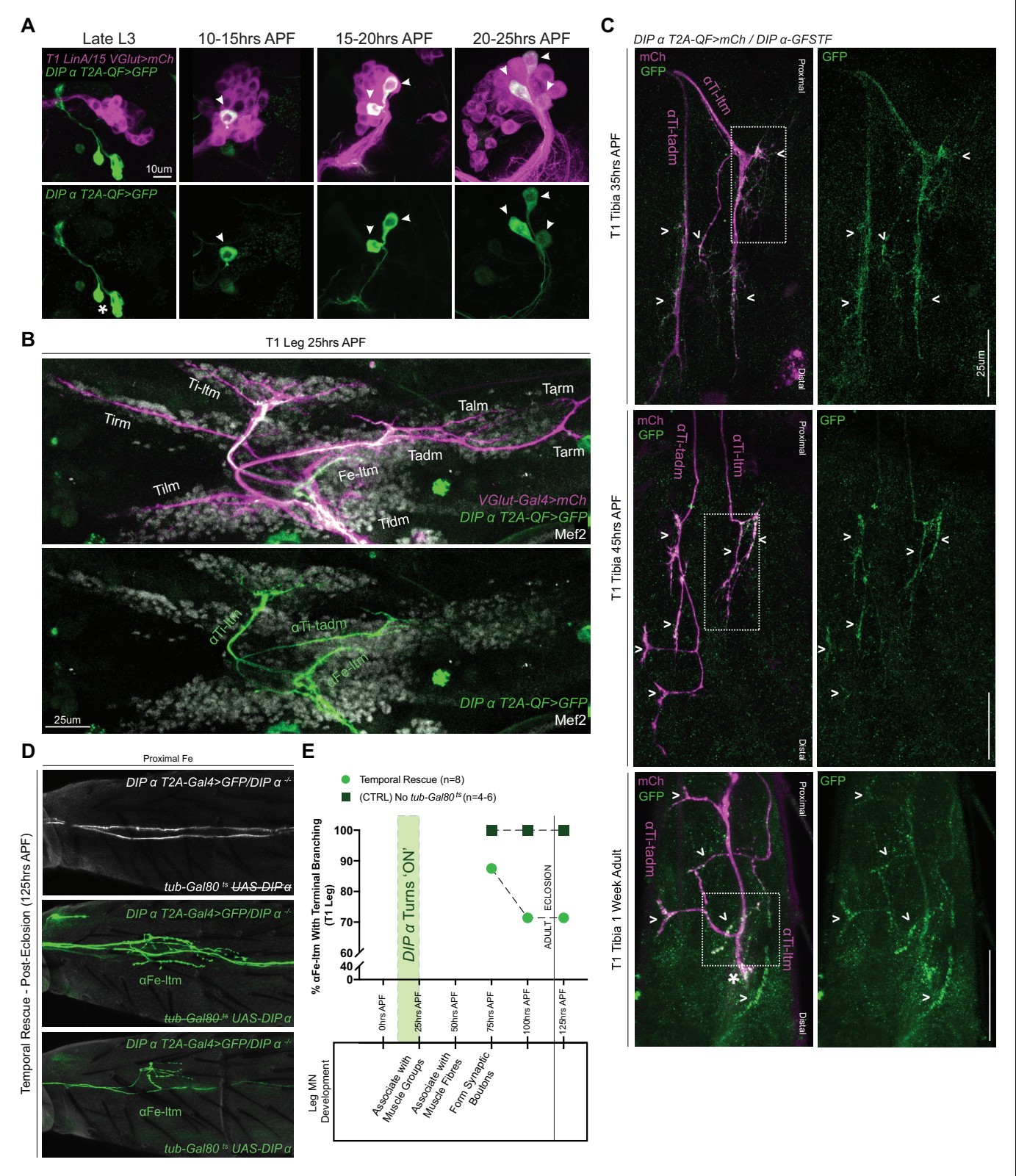

**Figure 4.** Spatial and temporal characterization of DIP-α expression. (**A**) T1 LinA/15 leg MN MARCM clones using *OK371-Gal4 > 20XUAS-6XmCherry* (magenta) and *DIP-α-T2A-QF > 10XQUAS-6XGFP* (green) to label leg MN cell bodies in the VNC at multiple developmental time points. At late L3 stages *DIP-α* expression is not yet 'ON' in LinA/15 leg MNs although expression is observed in non-LinA/15 cells (asterisk). Between 10–25 hr APF, the three LinA/15 α-leg MNs (*Figure 4—figure supplement 1A*) initiate DIP-α expression in a sequential manner, one after the other (arrowheads point to

*Figure 4 continued on next page*

*Figure 4 continued*

*DIP-α+cells in LinA/15 clones) (scale bar: 10 µm). (B) Pupal leg at 25 hr APF stained for all MNs (OK371-Gal4 > 20XUAS-6XmCherry; magenta), immature muscles (Mef2 expression; grey), and DIP-α-expressing MNs (green). (scale bar: 25 µm). (C) Endogenous DIP-α expression in αTi-ltm and αTi-tadm axon termini using GFP-tagged DIP-α-GFSTF (green, detected by anti-GFP, see Materials and methods) and labeled by DIP-α-T2A-QF > 10XQUAS-6XmCherry magenta) at 35 hr APF, 45 hr APF and in 1 week old adults. White arrowheads point to selected regions of mCherry and GFP co-expression. White-dotted boxes denote magnified insets in Figure 4—figure supplement 1D. (scale bar: 25 µm) (D) Temporal rescue at 125 hr APF of axon terminal branching of αFe-ltm in the proximal Fe of T1 adult legs in samples mutant for DIP-α using DIP-α-T2A-Gal4 > 20X-6XGFP, UAS-DIP-α-V5 and tub-Gal80^{ts} (see Supplementary File 2). Top row: Negative control (no UAS-DIP-α-V5) showing absence of αFe-ltm terminal branching in flies that were temperature shifted from 18°C to 30°C at 125 hr APF (axons, white; cuticle, grey). Middle row: Positive control (no tub-Gal80^{ts}) showing complete terminal branching of αFe-ltm in flies that were temperature shifted from 18°C to 30°C at 125 hr APF (axons, green; cuticle, grey). Bottom row: Temporal rescue of terminal branching of αFe-ltm in a DIP-α mutant background in flies that were temperature shifted from 18°C to 30°C at 125 hr APF; Terminal branches are shorter and/or fewer in number compared to the positive controls (axons, green; cuticle, grey). (E) Quantification of T1 leg samples with terminal branching of αFe-ltm in temporally rescued samples (N = 8) (green circles) compared to positive controls (N = 4–6) (no tub-Gal80^{ts}, dark green squares) that were temperature-shifted together at 75 hr, 100 hr or 125 hr APF. Terminal branching of αFe-ltm was seen in 87.5% of samples that were temperature-shifted at 75 hr APF and in 71.42% of samples that were temperature-shifted at 100 hr or 125 hr APF. Terminal branching was always observed in 100% of samples of the positive control and always absent in the negative control (N = 4–6, Figure 4C). Stages of leg MN axon development are indicated below the graph as defined in Figure 1. Initiation of endogenous DIP-α expression in the three WT LinA/15 α-leg MNs is indicated by a vertical green bar at 10 to 25 hr APF. Time of eclosion is indicated by a vertical line at 120 hr/5 days APF.*

DOI: https://doi.org/10.7554/eLife.42692.011

The following figure supplements are available for figure 4:

**Figure supplement 1.** Spatial and temporal characterisation of DIP-α expression.

DOI: https://doi.org/10.7554/eLife.42692.012

**Figure supplement 2.** Temporal rescue of DIP-α in a mutant background.

DOI: https://doi.org/10.7554/eLife.42692.013

Because DIP-α binds both Dpr6 and Dpr10 (*Carrillo et al., 2015*; *Özkan et al., 2013*), we next tested if expressing *dpr6* in muscles could rescue the terminal branching phenotypes of α-ltm MNs in *dpr10* mutants. Strikingly, using *Mef2-Gal4* to express Dpr6 in a *dpr10* mutant background we observed significant rescue in both αFe-ltm (88.8% of samples with terminal branching) and αTi-ltm (100% of samples with terminal branching) (*Figure 5B,D*).

In parallel to the above experiments, we also conducted an expression analysis of Dpr10, using a MiMIC-GFP protein-trap (*dpr10-GFSTF*) (*Nagarkar-Jaiswal et al., 2015*; *Tan et al., 2015*), during pupal development and observed that Dpr10 expression is 'ON' in subsets of muscle precursors in the leg imaginal discs at late L3 (*Figure 5—figure supplement 1D–E*) and remains expressed in adult leg muscles throughout pupal development (*Figure 5C*, *Figure 5—figure supplement 1F*). While Dpr10 was broadly observed in early pupal leg muscle precursors at 25 hr APF, we noticed higher levels in the ltms and depressor muscles in the Femur and Tibia at 45 hr APF (*Figure 5C*).

Taken together, the above results suggest that Dpr10 expression in the muscles normally interacts with DIP-α in a subset of leg MNs to ensure proper terminal branching of the DIP-α expressing leg MNs. Since exchanging the DIP-α binding partners, Dpr10 with Dpr6, in the muscles is sufficient to rescue terminal branching, we further conclude that the physical interaction, possibly adhesion, between leg MN axon termini and muscles provided by the DIP-Dpr interaction may be sufficient for the stereotyped terminal branches of these α-leg MNs.

## DIP-α is specifically required for terminal axon branching between 30 and 45 hr APF

Leg MN axons normally exhibit sequential rounds of defasciculation followed by dynamic branching during pupariation (*Figure 1*). The defects in terminal branching seen in DIP-α mutant leg MNs could potentially occur at any of the above stages. Since the MNs targeting the Ti-ltm show clear differences before and after defasciculation from secondary to tertiary bundles (*Figure 1*), we focused on characterizing terminal branching of the αTi-ltm in *DIP-α* mutants (*DIP-α-T2A-Gal4/DIP-α⁻*) compared to controls (*DIP-α-T2A-Gal4 > UAS-DIP-α*) using a combination of immunostaining and confocal imaging along with in vivo live imaging (*Supplementary file 2*). Although we focused these experiments on αTi-ltm, because it was more accessible to image than αFe-ltm, both MNs appear to behave similarly. Since mutant α-ltm MNs reach the vicinity of their muscle targets when examined in the adult, we expected the terminal branching defects in *DIP-α* mutants to occur after leg MNs

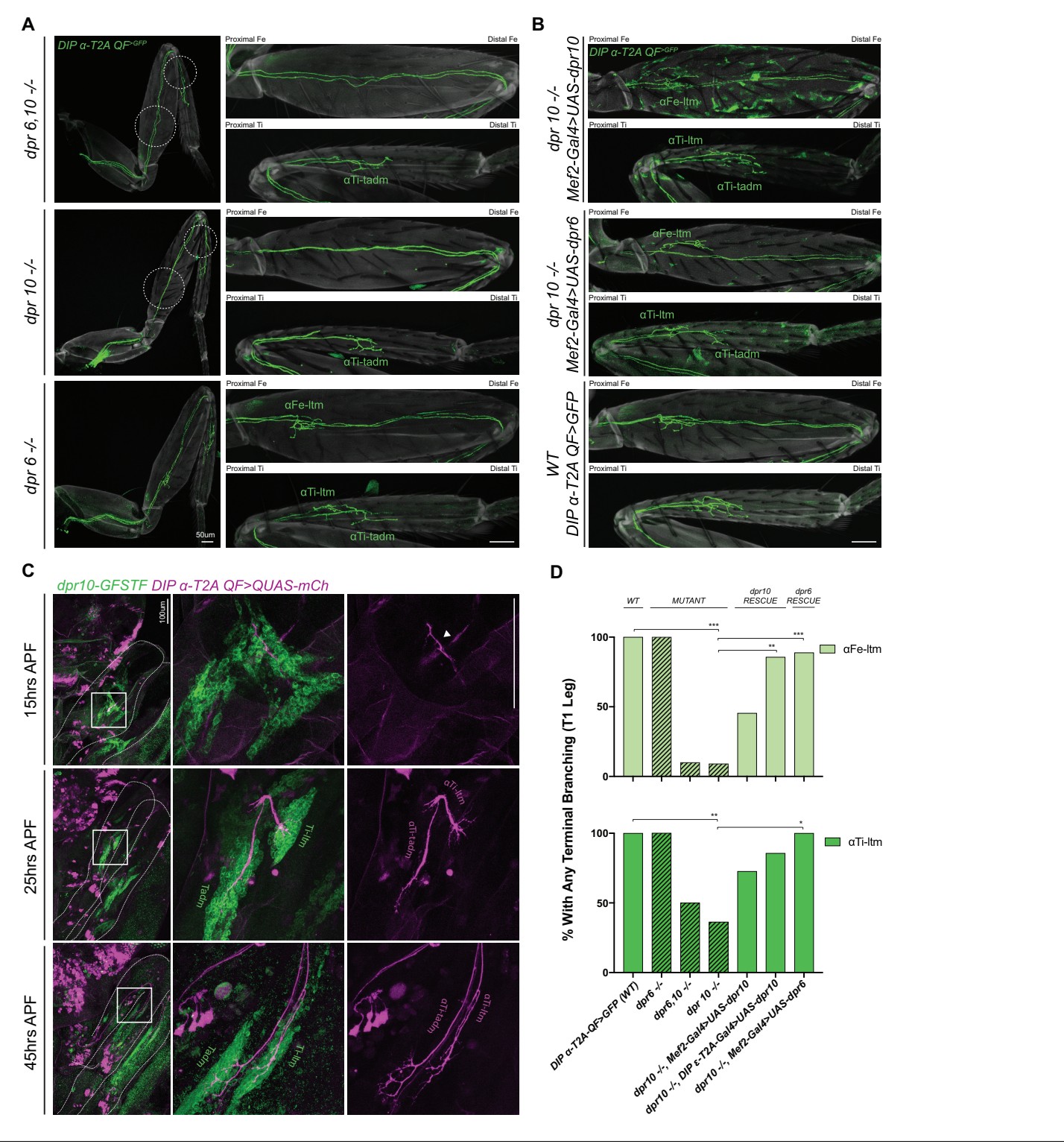

**Figure 5.** *dpr10* Expression in muscles is necessary for terminal branching of the α-leg MNs. (**A**) Terminal branching of the T1 α-leg MNs labeled by *DIP-α-T2A-QF > 10XQUAS-6XGFP* in *dpr6* and *dpr10* double and single mutants. Left – T1 legs; Right – Fe and Ti leg segments (axons, green; cuticle, grey). Terminal branching of αFe-ltm and αTi-ltm is absent only in the *dpr6, dpr10* double mutant and *dpr10* single mutant and phenocopies the *DIP-α* mutant phenotype (white dotted circles), while terminal branching of the α-leg MNs is intact in a *dpr6* single mutant. (scale bar: 50 μm) (**B**) Muscle-specific expression of *dpr10* (top) and *dpr6* (middle) using *Mef2-Gal4 > UAS-dpr10/6* V5 in a *dpr10* mutant background in Fe and Ti T1 leg segments showing rescue of terminal branching of αFe-ltm and αTi-ltm labeled by *DIP-α-T2A-QF > 10XQUAS-6XGFP*. Expression of *dpr10* in the muscles with

*Figure 5 continued on next page*

*Figure 5 continued*

the strong muscle driver, *Mef2-Gal4*, caused ectopic aberrant induction of *DIP-α-T2A-QF > 10XQUAS-6XGFP* expression in the cuticle of the leg. Wild-type terminal branching of the α-leg MNs is displayed using *DIP-α-T2A-QF > 10XQUAS-6XGFP* (bottom). (scale bar: 50 µm) (**C**) Endogenous *dpr10* expression in the developing T1 leg (Left column) using a GFP protein-trap inserted into a coding intron of *dpr10* (*Figure 5—figure supplement 1D*) (detected using a anti-GFP (green) – see Materials and methods) at 15 hr, 25 hr and 45 hr APF. Developing α-leg MNs are concurrently labeled by *DIP-α-T2A-QF > 10XQUAS-6XmCherry* (magenta) (middle column, merge; right column, *DIP-α-T2A-QF > 10XQUAS-6XmCherry*). At 15 hr APF (top row), at most only two of three α-leg MNs express *DIP-α* (immature axon terminals are indicated by a white arrowhead) and *dpr10* is broadly expressed in immature adult muscle precursors. By 25 hr APF (middle row) when axons are normally associated with their muscle groups, immature axons of αTi-ltm and αTi-tadm form filopodia in the *dpr10* expressing Ti-ltm and tadm. At 45 hr APF (bottom row) when leg MN axons are normally associated with distinct muscle fibers, αFe-ltm (*Figure 5—figure supplement 1F*), αTi-ltm and αTi-tadm have generated their terminal branches in the *dpr10* expressing Ti-ltm and Tadm. (scale bar: 100 µm) (**D**) Quantification of percentages of T1 leg samples with terminal branching of αFe-ltm (light green) and αTi-ltm (medium green) in WT (N = 15), mutant (diagonal lines) and rescue contexts (N = 7 to 11) using *dpr6* and *dpr10* double and single null mutations as indicated (see Materials and methods). Statistical significance was determined using Fisher's exact test.

DOI: https://doi.org/10.7554/eLife.42692.014

The following figure supplement is available for figure 5:

**Figure supplement 1.** *dpr10* expression in muscles is necessary for terminal branching of the α-Leg MNs.

DOI: https://doi.org/10.7554/eLife.42692.015

have sorted into their secondary axon bundles (20 to 25 hr APF). Indeed, when we image mutant samples at 25 hr APF along with a *VGlut* reporter to label all leg MNs, we see that mutant α-leg MNs, including αTi-ltm, properly sort into their secondary axon bundles (*Figure 6—figure supplement 1B*). Moreover, live in vivo imaging at 30 to 40 hr APF show that mutant αTi-ltm axons are also able to generate dynamic filopodia during leg extension (*Video 3*). However, by 30 hr APF branches innervating the developing Ti-ltm are shorter in length in mutant samples compared to the control (*Figure 6A*, *Figure 6—figure supplement 1A*). When we analyzed fixed samples between 30 to 50 hr APF, we noticed a gradual decrease in terminal branching in both αTi-ltm and αTi-tadm such that by 50 hr APF, mutant samples resemble the final adult phenotype (*Figure 6A*, *Figure 6—figure supplement 1B–C*). At this stage mutant αTi-ltm axons lack a prominent terminal branch and mutant αTi-tadm axons lack four proximal terminal branches and retain only the distal-most branch.

We next compared mutant and control samples at a slightly later time window, between ~35 and 45 hr APF, using live in vivo imaging (*Figure 6B*, *Videos 4–5*.). Control αTi-ltm samples initially generate several filopodial projections along the length of the axon terminal, which soon result in a stable terminal branch at the distal region of the main αTi-ltm axon. Although mutant αTi-ltms also generate filopodial projections, none of them result in the generation of a stable terminal branch. Instead, by ~45 hr APF, mutant αTi-ltm axons accumulate globular, punctate looking structures at their termini (*Figure 6—figure supplement 1B*). Defects in overall axon lengthening between mutant and control samples are also observed, with mutant samples terminating more proximally compared to control samples. The gradual decline in filopodial branching of the α-leg MNs in DIP-α mutants suggests that DIP-α is needed continuously between 30 and 45 hr APF to generate the correct length and number of terminal branches.

## Dpr10 protein is gradually restricted to distal fibers of the Ti-ltm 30 to 45 hr APF

From our live imaging analysis, we found that *DIP-α* is necessary to generate a stable terminal branch in αTi-ltm axons. However, DIP-α protein, which is localized along the entire αTi-ltm axon terminal during development (*Figure 4C*), cannot by itself explain the stereotyped terminal branch formation that occurs specifically at the distal region of the αTi-ltm axon. Therefore we took a closer look at Dpr10 protein expression in the Ti-ltm during development using an antibody against Dpr10 while simultaneously labeling developing muscles with Mef2 and the α-leg MNs with a GFP reporter (*Figure 7A–B*). At 25 hr APF Dpr10 is broadly observed in the entire immature Ti-ltm (*Figure 7A*) and does not specifically localize at positions of filopodial branch innervation (*Figure 7A′*). However, by 45 hr APF (*Figure 7B*), Dpr10 is enriched in the subset of distal Ti-ltm fibers that are targeted by the terminal branches of αTi-ltm MNs and is also highly concentrated at the precise locations of αTi-ltm branches (*Figure 7B′*). We also analyzed Dpr10 protein localization in animals where DIP-α was overexpressed (*DIP-α-T2A-Gal4 > UAS-DIP-α*) and observed a strong association between Dpr10

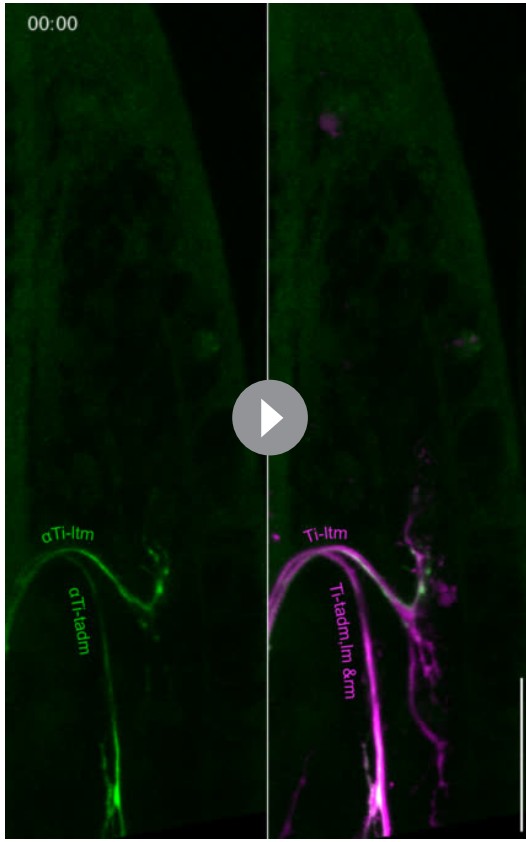

**Video 3.** Comparison of WT and DIP-α mutant Ti-ltm targeting leg MNs using live-imaging. Time lapse in vivo live imaging of Ti-ltm targeting leg MNs, including αTi-ltm, in a *DIP-α* mutant animal between ~30 hr APF (00:00) to ~38 hr APF (08:30) (10 min interval, five fps). Leg MNs are labeled using *VGlut-QF >10XQUAS-6xmCherry* (magenta) and αTi-ltm is labeled using *DIP-α-T2A-Gal4 > 20XUAS-6XGFP* (green) (left: αTi-ltm; right: Merge). In the first frame, both, the Ti-ltm and Ti-tadm, lm (levator muscles), and rm (reductor muscles) targeting secondary axon bundles (magenta), as well as individual αTi-ltm and αTi-tadm axons (green) within these bundles are visible. *DIP-α* mutant αTi-ltm axons generate dynamic filopodia during leg extension, but show a gradual decline in branching between 30 to 45 hr APF (*Figure 6A*, *Figure 6—figure supplement 1A–B*). (scale bar: 50 μm).
DOI: https://doi.org/10.7554/eLife.42692.018

expression and MN axon innervation at the Ti-ltm as early as 25 hr APF and up until 45 hr APF (*Figure 7—figure supplement 1B–C*). These results support the idea that DIP-α physically interacts with Dpr10 in vivo.

Interestingly, up until 35 hr APF, *DIP-α* mutant αTi-ltm axons still project filopodial branches towards Dpr10 expressing muscle precursor cells (*Figure 7—figure supplement 1A*). Since *DIP-α* mutant αTi-ltm axons display a gradual decline in terminal branching (*Figure 6*) these results suggest that while multiple mechanisms might be involved in directing filopodial branches towards their muscle targets, maintaining filopodial branching through DIP-α/Dpr10 interactions is required to promote additional branching, which together gradually refines the stereotyped terminal branching pattern.

## Discussion

In this study we used in vivo live imaging to describe the steps by which adult *Drosophila* leg MNs achieve their stereotyped axon terminal branch patterns at their muscle targets. By observing the process of leg MN targeting during pupariation, we began to query the relationships between the various steps, such as targeting the correct muscle, sequential axon defasciculation, organization of dynamic filopodial branches into stable terminal branches, fusion of muscle precursors into muscle fibers, and how these steps are ultimately coordinated with the morphogenesis of the adult leg with its complete proximo-distal axis.

We focused here on a small number of leg MNs and the role of the IgSF proteins, the DIPs and Dprs. Although many DIPs and Dprs are expressed in the adult neuromuscular system, we found a definitive requirement for DIP-α in MNs and one of its two cognate partners, Dpr10, in muscles for establishing the terminal branch pattern for three leg MNs. An analogous conclusion was made by examining phenotypes of the ISN-1s MN of the larva, suggesting a remarkably conserved role for this DIP-Dpr interaction at multiple stages of *Drosophila* neuromuscular development (see accompanying paper by *Ashley et al., 2018*). Moreover, we found that another DIP-α binding partner, Dpr6, which is not normally expressed in leg muscles, could functionally replace Dpr10 when expressed in muscles. As the amino acid residues in the interaction interface between DIP-α and Dpr6 are conserved in Dpr10 and are necessary for binding (*Carrillo et al., 2015*), these results suggest that binding between MN terminal branches and muscles, mediated by an extracellular protein-protein interaction, may be sufficient to establish the correct terminal branching pattern for these MNs. Additional evidence to support this idea comes from experiments in the *Drosophila* optic lobe where entirely heterologous interaction domains were used to replace

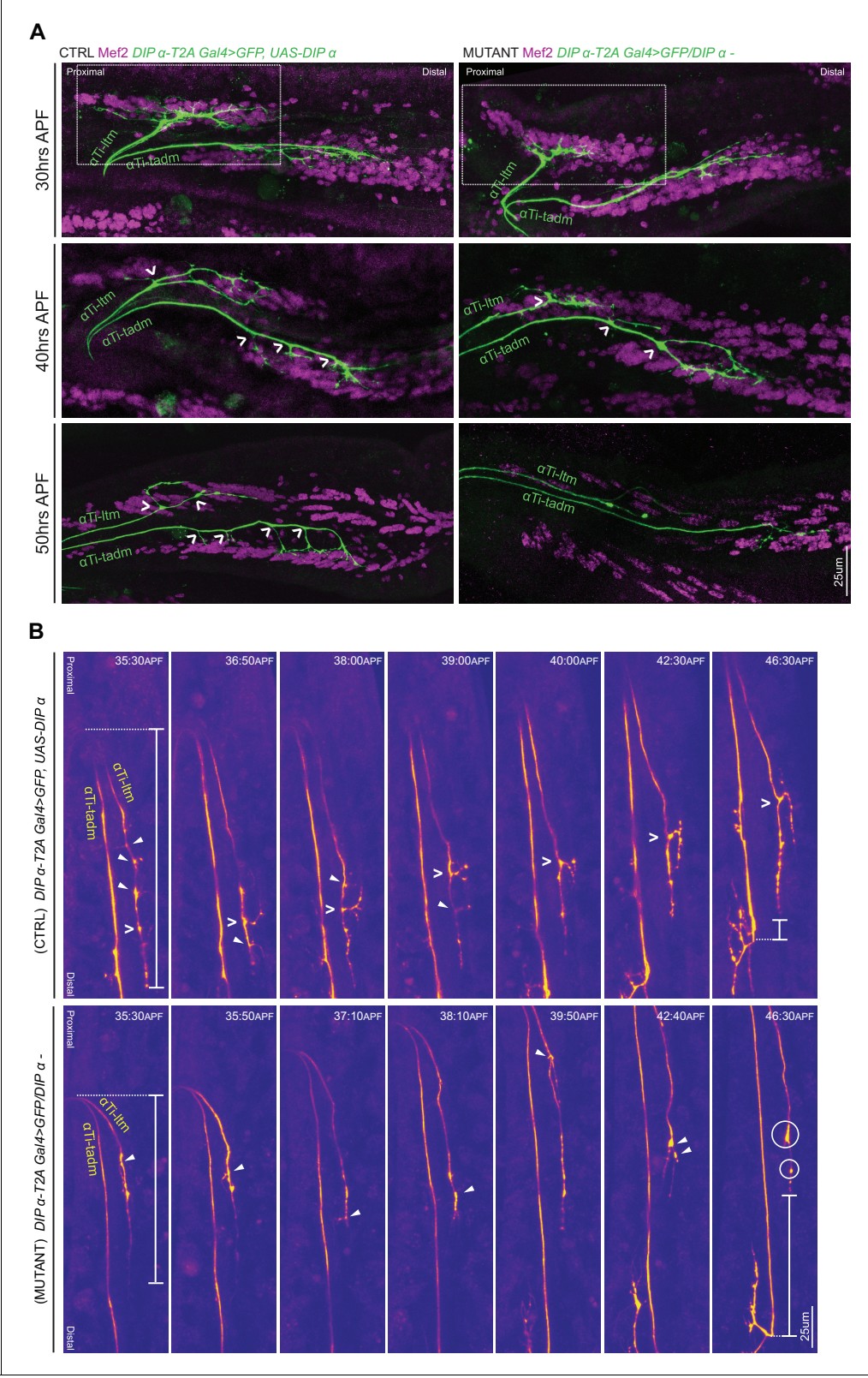

**Figure 6.** *DIP-α* is required for terminal axon lengthening and branching 30 to 45 hr APF. (**A**) Terminal axon branching of control (left) and *DIP-α* mutant (right) αTi-ltm and αTi-tadm leg MNs at 30 hr (top), 40 hr (middle) and 50 hr (bottom) APF using *DIP-α-T2A-Gal4 > UAS-DIP-α* and *DIP-α-T2A-Gal4/DIP-α⁻*, respectively. Axons are labeled using *DIP-α-T2A-Gal4 > 20XUAS-6XGFP* (green) and muscles are labeled with antibody against Mef2 (magenta). White arrowheads demarcate branch points along the axon terminal. At 50 hr APF, mutant αTi-ltm axons lack a prominent contralateral
*Figure 6 continued on next page*

*Figure 6 continued*

branch and mutant αTi-tadm axons lack four contralateral branches and retain the distal-most branch. White-dotted box denotes magnified inset in *Figure 5—figure supplement 1A* (scale bar: 25 µm). (B) Snapshots from time-lapse videos (*Video 4*, *Video 5*) comparing control (top) and mutant (bottom) αTi-ltm and αTi-tadm axons between ~35 hr and 45 hr APF (time-stamp is located on the top-right corner of each snapshot). Axons are labeled using *DIP-α-T2A-Gal4 > 20XUAS-6XGFP* (yellow). White open arrowheads demarcate the contralateral branch point on the αTi-ltm axon in the control sample while filled white arrowheads demarcate assorted dynamic filopodial projections along the αTi-ltm axon in both control and mutant samples. The distal-most tip of the αTi-ltm axon is more proximally located in the mutant sample compared to the control at ~35 hr APF (far left), as measured from the axon 'bend' at the joint between the distal Femur and proximal Tibia (denoted by white vertical bars) as well as at ~45 hr APF (far right), as measured from the distal most branch of αTi-tadm (denoted by white vertical bars). White circles demarcate globular punctate structures that form on the mutant αTi-ltm axon by ~45 hr APF (far right). (scale bar: 25 µm).
DOI: https://doi.org/10.7554/eLife.42692.016

The following figure supplement is available for figure 6:

**Figure supplement 1.** *DIP-α* is Required for terminal axon lengthening and branching 30 to 45 hr APF.
DOI: https://doi.org/10.7554/eLife.42692.017

extracellular DIP-α and Dpr10 interacting Ig domains and rescue a mutant phenotype (*Xu et al., 2018b*).

Notably, we found that neither *DIP-α* nor *dpr10* was required for MN axons to navigate to the correct muscle. However, the DIP-α–Dpr10 interaction appears to be critical to maintain the MN–muscle connection as the leg elongates and the muscles take their final shape. Based on these observations, we propose that α-leg MN axons target the correct cluster of muscle precursor cells during the first 20 hr of pupal development in a DIP/Dpr-independent manner, but then require this molecular interaction for the fine terminal branching pattern and for maintaining the MN–muscle interaction as the leg elongates and muscles mature to their final shape (*Figure 7C*). Interestingly, the transsynaptic cell adhesion complex comprising of Neurexin and Neuroligin is required for a similar process of terminal axon arbor growth in the abdominal body wall MNs in adult *Drosophila* (*Constance et al., 2018*) suggesting that multiple cell surface molecules are employed in different sub-cellular contexts to establish and maintain accurate terminal branching.

In general, the DIPs tend to be more restricted in their expression patterns compared to the Dprs in the leg neuromuscular system. The more limited expression patterns of DIPs has also been observed in other neural cell types (*Davis et al., 2018*; *Cosmanescu et al., 2018*), implying that differences in specificity and redundancy may be a general feature of these two Ig domain protein families. However, in contrast to *DIP-α*, we failed to observe obvious terminal branching or axon targeting defects for MNs that express other *DIP* genes, such as *DIP-γ* and *DIP-ζ*. One explanation for this observation is that *dpr10*, a strong binder of *DIP-α,* is unique among the *dpr* genes to be strongly expressed in leg muscles. Thus, it may be that other *DIPs* are playing roles in MN morphogenesis that are distinct from muscle targeting and terminal branching.

In addition to differences in how broadly the DIPs and Dprs are expressed, we also observed striking differences in the timing of their expression. Specifically, we found that Dpr10 begins to be expressed in leg muscle precursors as early as the late third instar larval stage (96 hr AEL). In contrast, DIP-α expression initiates in three leg MNs only after they have sorted into secondary axon bundles that subsequently associate with distinct muscle groups (15 to 25 hr APF). In *DIP-α* and *dpr10* mutants, α-leg MNs still sort into their secondary bundles but fail to establish terminal branches. Further, misexpressing DIP-α in non-α-expressing leg MNs as early as the late third instar stage had virtually no affect on their axon trajectories, consistent with the idea that these molecules are not involved in the initial steps of MN pathfinding. The initial broad expression pattern of Dpr10 in muscles might help promote early filopodial branching of the DIP-α expressing leg MNs while they are still fasciculated within their secondary bundles, thereby ensuring selective adhesion between the α-leg MN axons and their muscle partners during leg extension, a process that includes the physical rearrangement of muscle precursor cells into fibers. This is then followed by the gradual restriction of Dpr10 expression to specific muscle fibers and/or subregions on muscle fibers, which might contribute to the generation and stabilization of stereotyped terminal branching (*Figure 7C*). Both DIP-α and Dpr10 expression persist into the adult, and DIP-α localizes to pre-synaptic sites at mature NMJs (*Figure 4C*, *Figure 4—figure supplement 1D*), suggesting that this interaction might also be necessary for maintaining functional synapses. It is interesting to note, however, that muscle-

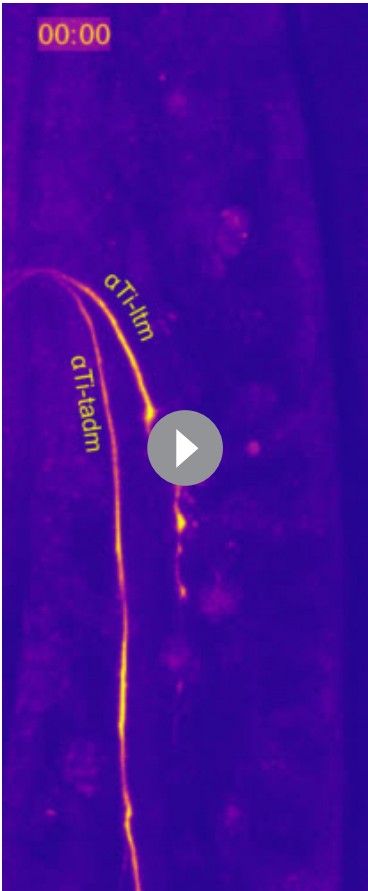

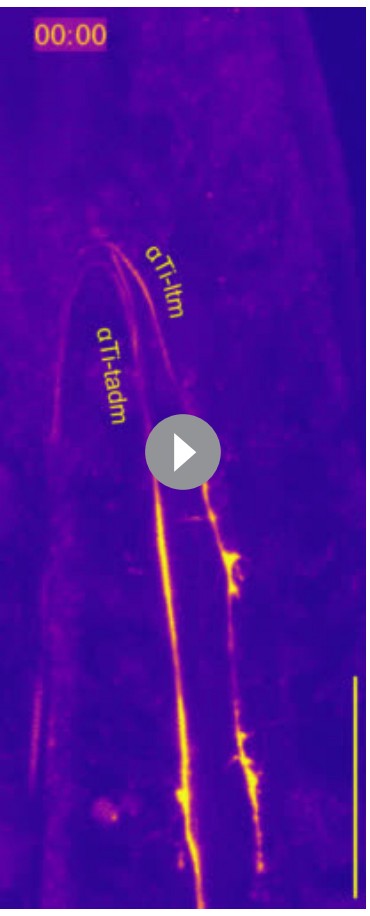

**Video 4.** Live imaging of WT αTi-ltm leg MN between ~35 hr APF to ~45 hr APF. Time lapse in vivo live imaging of αTi-ltm leg MNs in control (**Video 4**) and *DIP-α* mutant animal (**Video 5**), using *DIP-α-T2A-Gal4 > UAS-DIP-α* and *DIP-α-T2A-Gal4/DIP-α⁻* respectively, between ~35 hr APF (00:00) to ~45 hr APF (Control: 11:30; Mutant: 11:40) (10 min interval, five fps). α-leg MNs are labeled using *DIP-α-T2A-Gal4 > 20XUAS-6XGFP* (yellow). αTi-ltm and αTi-tadm axons are labeled in the first frame. The *DIP-α* mutant αTi-ltm axon fails to generate stable terminal branches, while the control αTi-ltm axon begins to generate a collateral branch at ~38 hr APF which stabilizes and extends in length by ~45 hr APF (**Figure 6B**). (scale bar: 50 μm).

DOI: https://doi.org/10.7554/eLife.42692.019

**Video 5.** Live imaging of *DIP-α* mutant αTi-ltm leg MN between ~35 hr APF to ~45 hr APF.

DOI: https://doi.org/10.7554/eLife.42692.020

specific rescue with Dpr10 was unable to recover branching in the larval MNs compared what we have observed in the adult (*Ashley et al., 2018*). We speculate that this might be due to the difference in the amount of time MNs have to establish their stereotyped branching in the larvae (several hours) and adult (several days), during which other cell-surface molecules involved in the branching process may have to be appropriately coordinated.

Interestingly, we observed consistent differences in the penetrance of the DIP-α and Dpr10 mutant phenotypes in the three leg MNs analyzed here. Terminal branching of αFe-ltm was lost in nearly every mutant sample. αTi-ltm, on the other hand, lost all of its terminal branches in only one-third of the mutant samples, with the remaining samples showing a partial loss of terminal branches. Finally, αTi-tadm only lost proximal terminal branches but always retained its distal most branch. Analogous to this latter phenotype, the DIP-α–Dpr10 interaction is also required for one of two terminal branches in the larval MN ISN-1s (see accompanying paper by *Ashley et al., 2018*). The decreasing dependencies of αFe-ltm, αTi-ltm and αTi-tadm on the DIP-α/Dpr10 interaction suggest that this interaction is context dependent. Interestingly, the number of tertiary bundles that these terminal branches stem from may be a relevant difference. αFe-ltm generates its terminal branches from a single tertiary bundle, while αTi-ltm does

so from two tertiary bundles, and the terminal branches of αTi-tadm stem from four distinct tertiary bundles (*Figure 6—figure supplement 1C*). Further, the targeted muscles also differ in their complexity: Fe-ltm comprises three muscle fibers, Ti-ltm comprises of six to seven fibers, and Ti-tadm is made up of twenty to twenty-four fibers in the foreleg (*Soler et al., 2004*). Therefore, as the morphological complexity of a MN and its muscle target increases, there may be a greater dependency on multiple molecular interactions, resulting in weaker phenotypes when only one interaction is removed. Consequently, we expect more combinations of interacting cell-surface proteins to function between leg MNs and muscles whose terminal branches stem from multiple tertiary bundles or have more complex muscle morphologies to navigate.

# Materials and methods

## Key resources table

| Reagent type (species) or resource | Designation | Source or reference | Identifiers | Additional information |
|---|---|---|---|---|
| Genetic reagent (*D. melanogaster*) | OK371-Gal4 | BDSC #26160 | RRID:BDSC_26160 | |
| Genetic reagent (*D. melanogaster*) | Vglut-T2A-QF2 | BDSC #60315 | RRID:BDSC_60315 | |
| Genetic reagent (*D. melanogaster*) | 10 C12-Gal4 | BDSC #47841 | RRID:BDSC_47841 | |
| Genetic reagent (*D. melanogaster*) | dpn > KDRT > Cre; Act > LoxP > LexA, LexA-myr::GFP; UAS-KD | PMID:24561995 | | |
| Genetic reagent (*D. melanogaster*) | Mef2-QF2 | BDSC #66469 | RRID:BDSC_66469 | |
| Genetic reagent (*D. melanogaster*) | 13XLexAop2-6XGFP | BDSC #52265 | RRID:BDSC_52265 | |
| Genetic reagent (*D. melanogaster*) | 10XQUAS-6XGFP | BDSC #52264; this paper | RRID:BDSC_52264 | VK0027 insertion generated for this paper. |
| Genetic reagent (*D. melanogaster*) | 10XQUAS-6XmCherry | BDSC #52269; BDSC #52270 | RRID:BDSC_52269; RRID:BDSC_52270 | |
| Genetic reagent (*D. melanogaster*) | 20XUAS-6X-GFP | BDSC #52261; BDSC #52262; This paper | RRID:BDSC_52261; RRID:BDSC_52262 | 86Fa insertion generated for this paper. |
| Genetic reagent (*D. melanogaster*) | 20XUAS-6X-mCherry | BDSC #52268 | RRID:BDSC_52268 | |
| Genetic reagent (*D. melanogaster*) | MiMIC-T2A-Gal4 lines | BDSC #7838; BDSC #76200; This paper | RRID:BDSC_78385; RRID:BDSC_76200 | Additional lines are listed in *Supplementary file 1*. generated by S.Nagarkar Jaiswal, H.Bellen and M.Courgeon, C. Desplan. |
| Genetic reagent (*D. melanogaster*) | DIP-α-A8-Gal4 | This paper | | attp2 and 86Fa insertions generated for this paper; See Materials and methods |
| Genetic reagent (*D. melanogaster*) | DIP-α-T2A-QF2 | This paper | | MiMIC Trojan Swaps generated for this paper; See Materials and methods |
| Genetic reagent (*D. melanogaster*) | DIP-β-T2A-QF2 | This paper | | MiMIC Trojan Swaps generated for this paper; See Materials and methods |
| Genetic reagent (*D. melanogaster*) | DIP-$\alpha^{1-7}$ also referred to as DIP-$\alpha^{null2}$ | PMID: 30467079 | | Generated by the Zipursky Lab |
| Genetic reagent (*D. melanogaster*) | DIP-$\beta^{1-95}$ | This paper | | Generated by the Zipursky Lab |
| Genetic reagent (*D. melanogaster*) | DIP-$\gamma^{1-67}$ | PMID: 30467079 | | Generated by the Zipursky Lab |

*Continued on next page*

Continued

| Reagent type (species) or resource | Designation | Source or reference | Identifiers | Additional information |
|---|---|---|---|---|
| Genetic reagent (D. melanogaster) | UAS-DIP-α-V5 | PMID: 30467079 | | Generated by the Zipursky Lab |
| Genetic reagent (D. melanogaster) | UAS-DIP-β | | | VK0027 insertion generated for this paper. |
| Genetic reagent (D. melanogaster) | dpr6,10⁻ also referred to as $dpr^{6-10L}$ | PMID: 30467079 | | Generated by the Zipursky Lab |
| Genetic reagent (D. melanogaster) | $dpr10^{1-29}$ also referred to as $dpr10^{null}$ | PMID: 30467079 | | Generated by the Zipursky Lab |
| Genetic reagent (D. melanogaster) | $dpr6^{1-116}$ also referred to as $dpr6^{null}$ | PMID: 30467079 | | Generated by the Zipursky Lab |
| Genetic reagent (D. melanogaster) | UAS-dpr10-V5 | PMID: 30467079 | | Generated by the Zipursky Lab |
| Genetic reagent (D. melanogaster) | UAS-dpr6-V5 | PMID: 30467079 | | Generated by the Zipursky Lab |
| Genetic reagent (D. melanogaster) | UAS-dpr10 RNAi VDRC | VDRC# 103511 | | |
| Genetic reagent (D. melanogaster) | tub > FRT .Gal80> | BDSC #38879; BDSC #38880 | RRID:BDSC_38879; RRID:BDSC_38880 | |
| Genetic reagent (D. melanogaster) | QUAS-DSCP-Flp0.2G (attp2) | BDSC #30008 | RRID:BDSC_30008 | |
| Genetic reagent (D. melanogaster) | $tub\text{-}Gal80^{ts}$ | BDSC #7108 | RRID:BDSC_7108 | |
| Genetic reagent (D. melanogaster) | DIP-α-GFSTF | BDSC #60523 | RRID:BDSC_60523 | |
| Genetic reagent (D. melanogaster) | dpr10-GFSTF | BDSC #59807 | RRID:BDSC_59807 | |
| Genetic reagent (D. melanogaster) | R13C09-Gal4 | BDSC #48555 | RRID:BDSC_48555 | |
| Genetic reagent (D. melanogaster) | hkb-Gal4 | BDSC #62578 | RRID:BDSC_62578 | |
| Genetic reagent (D. melanogaster) | DIP-ζ RNAi – TriP.HMS01671 | BDSC #38227 | RRID:BDSC_38227 | |
| Genetic reagent (D. melanogaster) | FRT42D | BDSC #1802 | RRID:BDSC_1802 | |
| Genetic reagent (D. melanogaster) | FRT42D tubG80, tubQS | PMID: 29395908 | | |
| Genetic reagent (D. melanogaster) | y,w,hs-Flp1.22, hs-Flp122.2(Chr 2) | Other | | Gift from Gary Struhl |
| Genetic reagent (D. melanogaster) | Mef2-Gal4 (Chr 2) | Other | | Bellen Lab, provided by R.Carrillo |
| Antibody | Rabbit polyclonal Anti-Mef2 | PMID: 7839146 | RRID:AB_2568604 | Generated by B.Paterson; (1:500) |
| Antibody | Sheep polyclonal Anti-GFP | Bio-Rad | Cat# 4745–1051 | (1:500) |
| Antibody | Chicken polyclonal Anti-GFP | Abcam | Cat# ab101863; RRID:AB_10710875 | (1:1000) |
| Antibody | Mouse monoclonal Anti-DIP-α | PMID: 30467079 | | Generated by the Zipursky Lab; (1:20) |
| Antibody | Mouse monoclonal Anti-Dpr10 | PMID: 30467079 | | Generated by the Zipursky Lab; (1:500) |
| Recombinant DNA reagent | T2A-QF2-Hsp70 | PMID:25732830 | RRID:Addgene_62944; RRID:Addgene_62945 | |

*Continued*

| Reagent type (species) or resource | Designation | Source or reference | Identifiers | Additional information |
|---|---|---|---|---|
| Recombinant DNA reagent | pJFRC28-10XUAS-IVS-GFP-p10 | PMID:22493255 | RRID:Addgene_36431 | |

Detailed fly genotypes are provided in *Supplementary file 2*.

## Temporal rescue of *DIP-α*

Two-day embryo collections were performed over a week at 18°C and since *Drosophila* develop at a slower rate at lower temperatures, external morphological features of the pupae were used to stage the flies (samples are referred to by their normal 25°C stage-time). Vials were then shifted together to 30°C for 5 days before dissection. Positive and negative controls were dissected along with experimental samples from each vial. Samples were included in the final analysis only when the positive controls displayed proper terminal branching.

## MARCM

To generate MARCM clones, embryos were collected for 12 hr at 25°C. First-instar larvae were heat shocked at 37°C for 25 mins. Adult progeny were screened under the fluorescent microscope for T1 clones.

## Adult leg and VNC dissection and mounting

Adult flies were first immersed in 80% Ethanol for ~1 min and rinsed in 0.3% PBT for ~15 mins. After removal of abdominal and head segments, adult legs attached to thoracic segments were fixed overnight at 4°C followed by atleast five washes in 0.3% PBT for 20 mins at room temperature. VNC and legs were dissected and mounted onto glass slides using Vectashield mounting medium (Vector Labs). Due to their large size, final leg images may be a composite of more than one image. Detailed protocol for leg dissection, mounting and imaging can be found in *Guan et al. (2018)*.

## Immunohistochemistry

### Antibodies

Rabbit Anti-Mef2 (1:500, Gift from B.Paterson), Sheep Anti-GFP (1:500, Biorad), Chicken Anti-GFP (1:1000, Abcam), Mouse Anti-DIP-α (1:20, Gift from S.L. Zipursky), Mouse Anti-Dpr10 (1:500, Gift from S.L. Zipursky), Rabbit Anti-Twist (1:300, Gift from K. Jagla), Mouse Anti-V5:549 (Biorad). Secondary antibodies used were Goat Anti-Rabbit Alexa 647 (Invitrogen); Goat Anti-Rabbit Alexa 555 (Invitrogen); Goat Anti-Guinea-pig Alexa 555 (Invitrogen); Goat Anti-Mouse Alexa 555 (Invitrogen); Donkey Anti-Mouse 647 (Jackson Immunolabs); Donkey Anti-Mouse 555 (Jackson Immunolabs, Gift from W.Grueber); Donkey Anti-Rabbit 555 (Jackson Immunolabs, Gift from W.Grueber); Donkey Anti-Sheep 488 (Jackson Immunolabs, Gift from C.Desplan); Goat Anti-Chicken Alexa 488 (Invitrogen)

### Dissections

Larval CNS and leg discs – Larvae were inverted to expose the CNS and attached leg imaginal discs; Adult VNC – After removal of the head, abdomen and legs, the thoracic ventral cuticle was removed to expose the adult VNC; Pupal legs – Pupae were extracted from the pupal case and dissected open from the dorsal surface along the A-P axis, followed by gentle washes with a 20 ul pipette to flush out the fat cells; Adult legs – T1 legs were dissected from the thoracic segment and transverse cuts were made across the middle of the Femur and Tibia segments with micro-dissection scissors.

### Immunostaining

Dissections were performed in 1XPBS, followed by fixation in 4% Formaldehyde (prepared with 1X PBS) for 25 mins or for 1 hr (adult legs) at room temperature. Samples were blocked for 2 hr (~3–5 washes) or overnight (adult legs) at room temperature and incubated with primary antibodies for one to two days and secondary antibodies for one day at 4°C. Fresh PBT with BSA (1XPBS, 0.3%

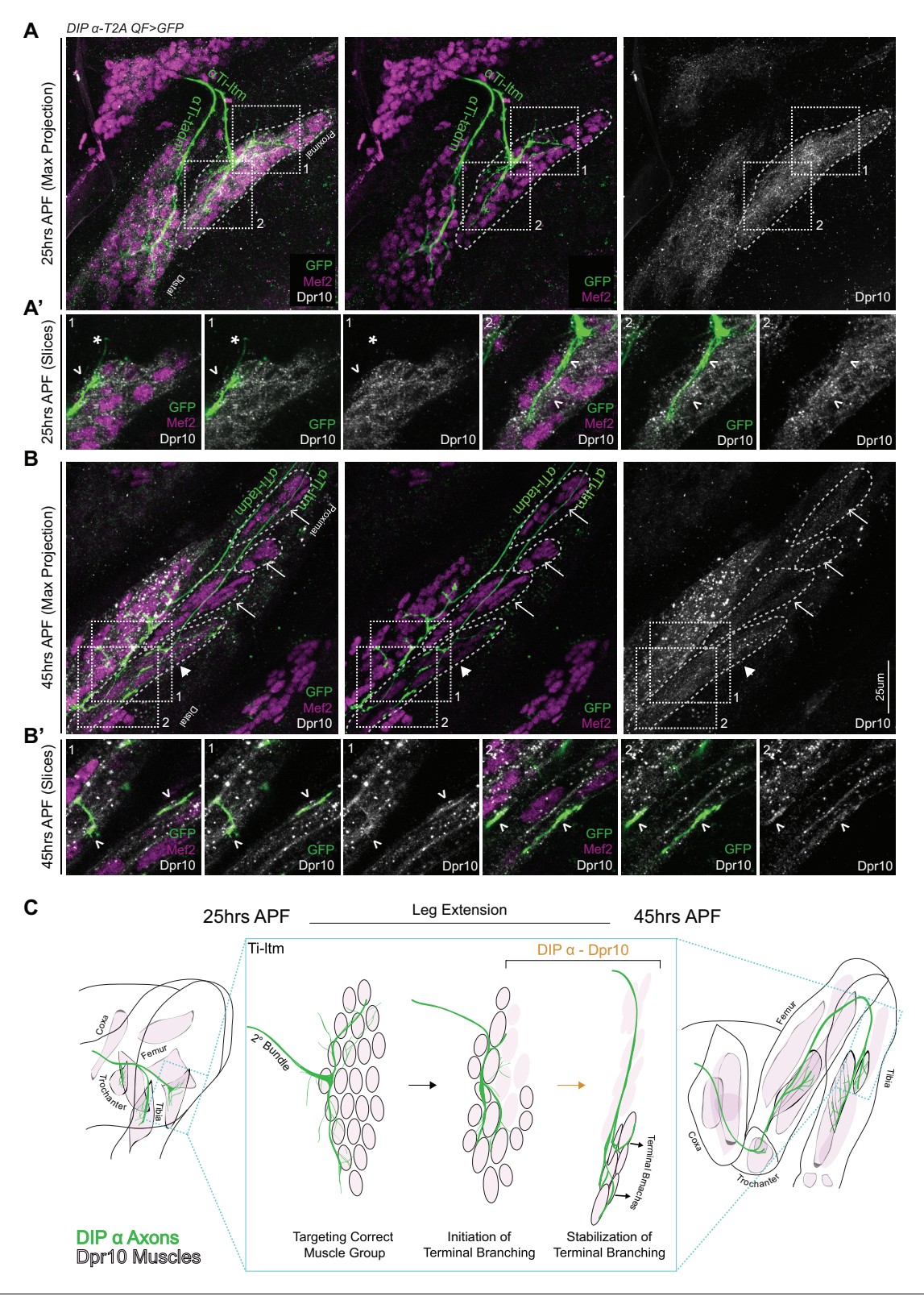

**Figure 7.** Dpr10 Expression is gradually restricted to distal fibers of the Ti-ltm 30 to 45 hr APF. (**A–B**) Dpr10 protein expression (grey) in the developing Ti-ltm and proximal tadm, labeled by Mef2 (magenta) along with αTi-ltm and αTi-tadm axons labeled by *DIP-α-T2AQF > 10XQUAS-6XGFP* (green) at 25 hr (**A**) and 45 hr (**B**) APF. Left: GFP, Mef2 and Dpr10; middle: GFP and Mef2; right: Dpr10. (**A′,B′**) show magnified single slice images of the dashed
*Figure 7 continued on next page*

Triton X-100, 1%BSA) was used for blocking, incubation and washing after fixation and after primary/secondary antibodies (~3–5 washes, 20 mins each). Samples were stored in Vectashield mounting medium (Vector Labs) until mounting and imaging.

### Mounting

Larval VNC and leg discs – Inverted larvae were cut along the body wall with micro-dissection scissors such that larval VNC and leg discs remained attached to each other and the body wall. Samples were mounted with VNCs oriented lateral side up; Adult VNC – VNC were dissected out from the thoracic segment and mounted ventral side up; Pupal legs – Pupae were mounted ventral side up; Adult legs – Adult leg segments were mounted lateral side up.

Samples were mounted in Vectashield mounting medium (Vectorlabs) on glass slides using sticker wells (iSpacer, SunJin Lab Co.).

## Microscopy

Multiple 0.5um-thick sections in the z-axis were imaged with a Leica TCS SP5 II. Binary images for z-stack images and 3D reconstructions were generated using Image J software (*Schneider et al., 2012*).

## Quantification and statistical analysis

For the binary quantification of the presence of terminal branching, T1 legs of multiple F1 animals obtained from parallel genetic crosses for each genotype were pooled together and scored for the presence of any amount of terminal branching in the leg MNs (sample size (N) is directly reported on the graph). Statistical significance was determined using Fisher's exact test and assigned using the following criteria: *$p<0.05$; **$p<0.01$; ***$p<0.001$.

For the quantification of branch number, automatic tracings of motor neurons from each genotype were obtained using Vaa3D (*Peng et al., 2010*; *Peng et al., 2014*) and the 'total number of tips' displayed in the 'morphology info' was used to calculate the branch number (sample size is reported on the graph). Data was assessed for normality using the Schapiro-Wilk normality test and statistical significance was determined using either a two-tailed unpaired t-test or a Mann-Whitney U test and assigned using the following criteria: *$p<0.05$; **$p<0.01$; ***$p<0.001$.

## In vivo live imaging

Pupae were first staged and sorted for the correct genotype. A small window on either the left/right ventral side of the pupal case was made using forceps to expose just the T1 leg. Individual pupae were placed on a glass slide, surrounded by two layers of filter paper dampened with distilled water. A 5 ul drop of distilled water was placed at the center of a glass coverslip (N-1.5) and placed exactly over the exposed T1 leg. Petroleum jelly surrounding the filter paper was used to seal the space between the coverslip and the glass slide to retain humidity. Samples were imaged on a Zeiss LSM700 microscope, 25X objective, with a 10 min interval between each z-stack series. Videos were generated using the FIJI software (*Schindelin et al., 2012*) at five frames per second.

## Plasmids and transgenic lines

*MiMIC-T2A-QF2* – Donor plasmids were obtained from Addgene (#62944 and #62945) and injected into BDSC stocks (#32808 and #34458 respectively). *Tan et al. (2015)* for detailed protocol. Transformants were screened and verified by crossing to *10XQUAS-6XGFP* (attp2).

*DIP-α-A8-Gal4* – Intronic region in the *DIP-α* genomic locus was PCR amplified from genomic DNA and inserted into a Gal4 vector with the DSCP promoter, generated by R.Voutev, Mann Lab and inserted into attp2 and 86Fa.

*DIP-α-A8* Forward Primer Nhe1: aatt<u>gctagc</u>cagtcgcaaaactcgttactcactc

*DIP-α-A8* Reverse Primer AgeI: aatt<u>accggt</u>aagatattaaaaaacatcaggaattatttctctc

*UAS-DIP-β - DIP-β* cDNA (synthetically generated and provided by S.L.Zipursky) was PCR amplified and inserted into pJFRC28 (Addgene #36431) using Not1 and Xba1. Plasmid DNA was inserted into VK00027.

Hexameric Fluorescent reporters – Original plasmids were obtained from S.Stowers and inserted into VK00027 and 86fa.

Injection services were provided by BestGene Inc.

## Acknowledgements

We thank SL Zipursky for generously sharing fly stocks and reagents; M Courgeon and C Desplan for sharing *MiMIC-T2A-Gal4* lines, antibodies and other fly stocks and reagents; V Fernandes for help with the in vivo live imaging; M Eveland for identifying *hkb-Gal4* expression in LinB/24 leg MNs; G Struhl, W Grueber and M Kohwi and their lab members for sharing time on their confocal microscopes; Members of the Mann lab, W Grueber, O Hobert, R Carrillo, and K Zinn for comments and suggestions. This work was funded by NIH grants R01NS070644 and U19NS104655 to RSM and R01GM067858 to H Bellen.

## Additional information

### Funding

| Funder | Grant reference number | Author |
| --- | --- | --- |
| National Institutes of Health | R01NS070644 | Richard S Mann |
| National Institutes of Health | U19NS104655 | Richard S Mann |

The funders had no role in study design, data collection and interpretation, or the decision to submit the work for publication.

### Author contributions

Lalanti Venkatasubramanian, Conceptualization, Data curation, Formal analysis, Validation, Investigation, Visualization, Methodology, Writing—original draft, Writing—review and editing; Zhenhao Guo, Resources, Generation of DIP-Alpha-A8-Gal4; Shuwa Xu, Resources, Generated DIP/dpr reagents; Liming Tan, Qi Xiao, Resources; Sonal Nagarkar-Jaiswal, Resources, Generated MiMIC-T2A-Gal4 lines; Richard S Mann, Conceptualization, Supervision, Funding acquisition, Methodology, Writing—original draft, Writing—review and editing

### Author ORCIDs

Lalanti Venkatasubramanian https://orcid.org/0000-0002-9280-8335
Richard S Mann http://orcid.org/0000-0002-4749-2765

### Decision letter and Author response

Decision letter https://doi.org/10.7554/eLife.42692.027
Author response https://doi.org/10.7554/eLife.42692.028

## Additional files

### Supplementary files

• Supplementary file 1. *DIP* and *dpr MiMIC-T2A-Gal4* lines .
DOI: https://doi.org/10.7554/eLife.42692.023
• Supplementary file 2. Genotypes used for each figure.
DOI: https://doi.org/10.7554/eLife.42692.024
• Transparent reporting form
DOI: https://doi.org/10.7554/eLife.42692.025

### Data availability

All data generated or analysed during this study are included in the manuscript and supporting files.

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
