## [Decision Letter]

Thank you for submitting your article "Stereotyped Terminal Axon Branching of Leg Motor Neurons Mediated by IgSF Proteins DIP-α and Dpr10" for consideration by *eLife*. Your article has been reviewed by K VijayRaghavan as the Senior Editor, a Reviewing Editor, and three reviewers. The following individual involved in the review of your submission has agreed to reveal her identity: Durafshan Sakeena Syed (Reviewer #1).

The reviewers have discussed the reviews with one another and the Reviewing Editor has drafted this decision to help you prepare a revised submission.

Summary:

This work provides one of the first examples of regulatory cues that operate in developing neuromuscular systems to ensure proper innervation of a specific multifibrillar muscle and the formation of a stabilized NMJs. This is a particularly well-illustrated study that uses large sets of up-to-date genetic tools (and their combinations) allowing precise mapping of neural networks and gene expression. It is worth mentioning that performing such an analysis in the context of developing *Drosophila* leg is from a technical point of view highly challenging and the quality of presented data including in vivo imaging and precise mapping of Dpr10 protein expression during development of tibia and femur leg muscles are excellent. The interpretations of complex patterns and phenotypes are accurate. The manuscript is of high importance for the field.

It is a strength that the introduction combines conclusions drawn from vertebrates and invertebrates, but given the substantial differences, the authors should note which is which.

Essential revisions:

1) Authors suggest that the terminal axonal branching establishes between 25-45 hours APF. Although by 45 hours APF, three distinct tertiary bundles of the Ti-ltm motor neurons are resolved, the stereotyped terminal innervation pattern does not seem to be finalized yet. Live imaging data (Video 2) indicates that the dynamic branches and growth cones are still present until the end of the movie (40-50 hours APF). And the terminal arborization in adults does not look similar to 45 hours APF (Figure 1—figure supplement 1). Also, seen in Brierley et al., (2011), axons continue to elaborate their terminals from 30-60 hours APF, and by 70 hours APF the morphology is indistinguishable from that of an adult.

If this is true, then further elaboration of terminal branches, axonal pruning, synapse formation, and stabilization might happen around 45-60 hours APF. It would be useful if synaptic marker and muscle membrane reporter (rather than nuclear Dmef2) are used to study the developmental profile. Motor neurons expressing GFP and the synaptic marker rab3::YFP (Enriquez et al.,) under the control of VGlut/lin15/DIP-α driver, and Dmef2>mCD8-RFP (or MHC-RFP/ Phalloidin after 35 hours APF) would reveal contact points and interaction between the terminal branches and muscle membrane, and the timing of the synapse formation and stabilization.(This is not a required experiment: Please discuss the point raised, if the results are not already available or the experiments are not speedily doable.)

2) Authors show that the DIP-α is specifically required for terminal axon branching between 30 and 45 hours APF. Although authors indicate that the filopodia are extended and lost and are unable to form stable connections around 30-45 hours APF in DIP-α mutants, this does not rule out its role in elongation, pruning, synapse formation, and stabilization after 45 hours APF. It could be easily tested by using DIP-α RNAi with VGlut or lin15 or DIP-α driver (and Dpr10 RNAi with Dmef2 or MHC driver) at different developmental stages (i) from 45-60 hours APF to see if it affects pruning and synapse formation, (ii) knockdown after eclosion to study if it plays a role in synaptic maintenance. In the null mutant, although the earliest phenotype is evident, the latter role will be missed. Also, the fact that authors are able to rescue the phenotype till late stages of development point to these multiple roles in achieving stereotyped terminal arborization. (This is not a required experiment: Please discuss the point raised, if the results are not already available or the experiments are not speedily doable.)

3) Authors test whether DIP-α is sufficient to induce terminal branching of ltms on MNs that normally do not target these muscles by ectopically expressing DIP-α in lin24 MNs. It provides another evidence that DIP-α is not required for targeting, but it does not indicate its role in terminal arborization. Since all the motor neurons that innervate ltm do not seem to express DIP-α, it will be interesting to test if the ectopic DIP-α expression in these MNs changes their terminal arborization. This could be tested by using sparse labeling ltm MN lines /MARCM or DIP-β Ti-ltm driver (used in Figure 3—figure supplement 2) to ectopically express DIP-α. (This is not a required experiment: Please discuss the point raised, if the results are not already available or the experiments are not speedily doable.)

4) Authors show that the Dpr10 expression in muscle precursors starts earlier than the DIP-α expression in the motor neurons. Also, Drp10 rescue experiment in muscles (Figure 5) induced ectopic expression of DIP-α in additional cells. Does this suggest that Dpr10 induces DIP-α expression and clustering? Is the temporal rescue of Dpr10 similar to that of DIP-α?

5) Finally, is the specific terminal branching pattern of motor neurons important for coordinated movement? Since both DIP-α and Dpr10 mutants show defects in axonal branching, do these flies show walking defects?

6) Figure 2A and Figure 2—figure supplement 1A are quite unhelpful. The many panels (a total of 18) are nearly indistinguishable even though they are meant to show distinct patterns. Figure 2—figure supplement 1C is more informative and perhaps could be included in a main figure.

7) Subsection “DIP-α is Necessary for the Terminal Branching of Three Leg MNs”: Names of the relevant motor neurons and muscles seem to be inconsistent and end up confusing a non-expert reader. It would be helpful to choose a single, simple name for each, introduce it early, and use it throughout.

8) Subsection “DIP-α is Necessary for the Terminal Branching of Three Leg MNs”: Lack of a gain of function phenotype contrasts with results from Ashley et al. Is there evidence that the ectopically expressed protein reaches the surface? Also, since the authors are clearly aware of each other's work, it would be helpful to discuss this difference.

9) Subsection “Dpr10 Expression in Muscles is Necessary for Terminal Branching of the α-Leg MNs”: It seems that dpr6 and dpr10 aren't normally redundant but that overexpression of dpr6 in cells that express it anyway can rescue a dpr10 mutant phenotype. This is odd. Does it result from high levels of overexpression with perhaps lower affinity of dpr6 than drp10 for the DIP? Some discussion is necessary.

10) In a few places, the authors mention that DIP expression is more restricted than dpr expression. If they want to emphasize this point, they should cite previous literature (including papers cited as in press) and make some statement in the discussion on whether this is a generalizable conclusion.

---

## [Author Response]

Summary:This work provides one of the first examples of regulatory cues that operate in developing neuromuscular systems to ensure proper innervation of a specific multifibrillar muscle and the formation of a stabilized NMJs. This is a particularly well-illustrated study that uses large sets of up-to-date genetic tools (and their combinations) allowing precise mapping of neural networks and gene expression. It is worth mentioning that performing such an analysis in the context of developing Drosophila leg is from a technical point of view highly challenging and the quality of presented data including in vivo imaging and precise mapping of Dpr10 protein expression during development of tibia and femur leg muscles are excellent. The interpretations of complex patterns and phenotypes are accurate. The manuscript is of high importance for the field.It is a strength that the introduction combines conclusions drawn from vertebrates and invertebrates, but given the substantial differences, the authors should note which is which.

We have revised the text accordingly.§

Essential revisions:1) Authors suggest that the terminal axonal branching establishes between 25-45 hours APF. Although by 45 hours APF, three distinct tertiary bundles of the Ti-ltm motor neurons are resolved, the stereotyped terminal innervation pattern does not seem to be finalized yet. Live imaging data (Video 2) indicates that the dynamic branches and growth cones are still present until the end of the movie (40-50 hours APF). And the terminal arborization in adults does not look similar to 45 hours APF (Figure 1—figure supplement 1). Also, seen in Brierley et al., (2011), axons continue to elaborate their terminals from 30-60 hours APF, and by 70 hours APF the morphology is indistinguishable from that of an adult.If this is true, then further elaboration of terminal branches, axonal pruning, synapse formation, and stabilization might happen around 45-60 hours APF. It would be useful if synaptic marker and muscle membrane reporter (rather than nuclear Dmef2) are used to study the developmental profile. Motor neurons expressing GFP and the synaptic marker rab3::YFP (Enriquez et al.,) under the control of VGlut/lin15/DIP-α driver, and Dmef2>mCD8-RFP (or MHC-RFP/ Phalloidin after 35 hours APF) would reveal contact points and interaction between the terminal branches and muscle membrane, and the timing of the synapse formation and stabilization.(This is not a required experiment: Please discuss the point raised, if the results are not already available or the experiments are not speedily doable.)

After 45 hours APF tertiary bundles stay intact and primary ‘branches’ are established. Certainly branching, elaboration and pruning continues to occur until 60 hours APF. We have reworded the text to indicate this. Live imaging with muscles and a synaptic marker will be technically challenging, as we require strong drivers and multiple copies of the reporter transgenes. Furthermore, fluorescent reporters when driven with *Mef2-Gal4* result in expression in the fat bodies, which obstructs the view of the MNs. Since this publication is primarily focused on the DIP-Α/Dpr10 interaction, we would like to first optimize the live-imaging tools to put together a separate publication to emphasize the live-imaging technique for adult leg MNs.

2) Authors show that the DIP-α is specifically required for terminal axon branching between 30 and 45 hours APF. Although authors indicate that the filopodia are extended and lost and are unable to form stable connections around 30-45 hours APF in DIP-α mutants, this does not rule out its role in elongation, pruning, synapse formation, and stabilization after 45 hours APF. It could be easily tested by using DIP-α RNAi with VGlut or lin15 or DIP-α driver (and Dpr10 RNAi with Dmef2 or MHC driver) at different developmental stages (i) from 45-60 hours APF to see if it affects pruning and synapse formation, (ii) knockdown after eclosion to study if it plays a role in synaptic maintenance. In the null mutant, although the earliest phenotype is evident, the latter role will be missed. Also, the fact that authors are able to rescue the phenotype till late stages of development point to these multiple roles in achieving stereotyped terminal arborization. (This is not a required experiment: Please discuss the point raised, if the results are not already available or the experiments are not speedily doable.)

We are very interested in identifying whether DIP-Αlpha plays a role in synaptic formation/stabilization. However, we are unlikely to have these data soon enough and we have addressed this interesting point in the Discussion section.

3) Authors test whether DIP-α is sufficient to induce terminal branching of ltms on MNs that normally do not target these muscles by ectopically expressing DIP-α in lin24 MNs. It provides another evidence that DIP-α is not required for targeting, but it does not indicate its role in terminal arborization. Since all the motor neurons that innervate ltm do not seem to express DIP-α, it will be interesting to test if the ectopic DIP-α expression in these MNs changes their terminal arborization. This could be tested by using sparse labeling ltm MN lines /MARCM or DIP-β Ti-ltm driver (used in Figure 3—figure supplement 2) to ectopically express DIP-α. (This is not a required experiment: Please discuss the point raised, if the results are not already available or the experiments are not speedily doable.)

We have done this experiment with some tibia targeting MN-specific Gal4 lines and do not see an obvious difference in branching (i.e. shift from proximal Ti-ltm to distal Ti-ltm). We also do not see a noticeable increase in branching in the femur or tibia when we express *DIP-Α* in LinA MARCM clones however this is hard to verify given the number of MNs neurons labeled (~30). We prefer not to include these data in the manuscript, because this is not a main point of the paper and requires verifying the Gal4-lines to make sure that they are indeed strong drivers.

4) Authors show that the Dpr10 expression in muscle precursors starts earlier than the DIP-α expression in the motor neurons. Also, Drp10 rescue experiment in muscles (Figure 5) induced ectopic expression of DIP-α in additional cells. Does this suggest that Dpr10 induces DIP-α expression and clustering? Is the temporal rescue of Dpr10 similar to that of DIP-α?

We have not attempted the temporal rescue for *Dpr10* due to technical difficulties. It might be that high ectopic expression of *Dpr10* with *Mef2-Gal4* is capable of inducing expression of the MiMIC.

5) Finally, is the specific terminal branching pattern of motor neurons important for coordinated movement? Since both DIP-α and Dpr10 mutants show defects in axonal branching, do these flies show walking defects?

We have not attempted this experiment, as we would be perturbing the terminal branching of only three motor neurons and it would be difficult to correlate behavioral and MN-targeting defects in whole-animal mutants. Furthermore, tertiary bundles can comprise of terminal branching from multiple MNs so ideally, we would like to eliminate specific tertiary bundles (maybe through laser ablation or induced cell-death) and then perform the behavioral analysis. Although interesting, these experiments are, we believe, outside the scope of this manuscript.

6) Figure 2A and Figure 2—figure supplement 1A are quite unhelpful. The many panels (a total of 18) are nearly indistinguishable even though they are meant to show distinct patterns. Figure 2—figure supplement 1C is more informative and perhaps could be included in a main figure.

Figure 2—figure supplement 1A shows the similarity in *dpr* expression but we have included Figure 2—figure supplement 1C in the main figure as Figure 2E.

7) Subsection “DIP-α is Necessary for the Terminal Branching of Three Leg MNs”: Names of the relevant motor neurons and muscles seem to be inconsistent and end up confusing a non-expert reader. It would be helpful to choose a single, simple name for each, introduce it early, and use it throughout.

To avoid confusion, we have made sure we were consistent with the nomenclature to refer to these MNs throughout the manuscript. Because there are ~50 MNs targeting each leg, we chose to refer to the relevant MNs by both their DIP-a expression and which muscle they target, e.g. αFe-ltm expresses DIP-a and targets the ltm muscle in the femur while αTi-ltm expresses DIP-a and targets the ltm muscle in the tibia. We are also careful to define these early in the manuscript and mention other names at that time to avoid confusion (e.g.Fe-ltm (muscle) has previously been referred to as ltm2 in the literature). We feel that this is the clearest way to refer to them.

8) Subsection “DIP-α is Necessary for the Terminal Branching of Three Leg MNs”: Lack of a gain of function phenotype contrasts with results from Ashley et al. Is there evidence that the ectopically expressed protein reaches the surface? Also, since the authors are clearly aware of each other's work, it would be helpful to discuss this difference.

V5 staining in the supplementary figure shows *Mef2>UAS-dpr10* at the surface of muscle cells at 25 hours APF. We did recover partial rescue with *Mef2-Gal4* and one possible reason we saw any rescue compared to Ashley et al., was because terminal branching occurs over a larger time-scale in comparison to the larvae — we have updated the Discussion accordingly.

9) Subsection “Dpr10 Expression in Muscles is Necessary for Terminal Branching of the α-Leg MNs”: It seems that dpr6 and dpr10 aren't normally redundant but that overexpression of dpr6 in cells that express it anyway can rescue a dpr10 mutant phenotype. This is odd. Does it result from high levels of overexpression with perhaps lower affinity of dpr6 than drp10 for the DIP? Some discussion is necessary.

Firstly, to clarify, Dpr6 is NOT normally expressed in the leg muscles. We believe that binding specificity between the DIP-dpr partners is sufficient to rescue the mutant phenotype even in the ectopic context. We discuss this and mention the nectin-nectin rescue from the recent Zipursky paper.

10) In a few places, the authors mention that DIP expression is more restricted than dpr expression. If they want to emphasize this point, they should cite previous literature (including papers cited as in press) and make some statement in the discussion on whether this is a generalizable conclusion.

The recent single-cell RNA-Seq paper of the adult brain and the new DIP-dpr interactome paper from the Shapiro and Honig labs, which we have now referenced, both suggest that dprs are much more widely expressed than the DIPs.